# A method for high-throughput production of sequence-verified DNA libraries and strain collections

Justin D Smith[1,2,†] , Ulrich Schlecht[1,3,†], Weihong Xu[1,4,†], Sundari Suresh[1,3], Joe Horecka[1,3], Michael J Proctor[1,3], Raeka S Aiyar[1,3] , Richard A O Bennett[5,6], Angela Chu[1,3], Yong Fuga Li[1], Kevin Roy[1,2], Ronald W Davis[1,2,3], Lars M Steinmetz[1,2,7], Richard W Hyman[1,3], Sasha F Levy[5,6] & Robert P St.Onge[1,3,*]

## Abstract

The low costs of array-synthesized oligonucleotide libraries are empowering rapid advances in quantitative and synthetic biology. However, high synthesis error rates, uneven representation, and lack of access to individual oligonucleotides limit the true potential of these libraries. We have developed a cost-effective method called Recombinase Directed Indexing (REDI), which involves integration of a complex library into yeast, site-specific recombination to index library DNA, and next-generation sequencing to identify desired clones. We used REDI to generate a library of ~3,300 DNA probes that exhibited > 96% purity and remarkable uniformity (> 95% of probes within twofold of the median abundance). Additionally, we created a collection of ~9,000 individually accessible CRISPR interference yeast strains for > 99% of genes required for either fermentative or respiratory growth, demonstrating the utility of REDI for rapid and cost-effective creation of strain collections from oligonucleotide pools. Our approach is adaptable to any complex DNA library, and fundamentally changes how these libraries can be parsed, maintained, propagated, and characterized.

**Keywords** arrayed strain collection; CRISPR interference; DNA libraries; oligonucleotide pools; synthetic biology

**Subject Categories** Chromatin, Epigenetics, Genomics & Functional Genomics; Methods & Resources

**Mol Syst Biol. (2017) 13: 913**

## Introduction

Array-based DNA synthesis can produce short DNA oligonucleotides at larger scales and lower costs than column-based platforms (Kosuri & Church, 2014). While array-synthesized oligonucleotides are useful for a variety of applications, including gene synthesis, genome editing, functional genomics, and molecular detection, they are provided as complex mixtures, can be unevenly represented, and contain a significant number of synthesis-derived errors (Klein et al, 2015). The probability of at least one synthesis error increases with the addition of each base during oligonucleotide synthesis. For example, a commonly observed error frequency of 1 in 200 will result in < 37% of 200-mer DNA being of correct sequence (calculated as $0.995^{200}$). A growing need for inexpensive synthetic DNA has motivated strategies for improving the utility of these oligonucleotide libraries (Kim et al, 2012; Schwartz et al, 2012; Lee et al, 2015). Here, we describe Recombinase Directed Indexing (REDI), which utilizes site-specific recombination to index (i.e. barcode) DNA libraries, thereby enabling the sequencing and subsequent high-throughput retrieval of clonal DNA. The REDI approach provides a mechanism for generating uniform, sequence-verified DNA libraries of varying composition on demand. Additionally, REDI allows for access to, and rapid cell-based functional interrogation of individual library members. The methods presented here are not limited to use with oligonucleotide libraries but are extendable to other types of DNA libraries (e.g. existing CRISPR and shRNA functional genomic libraries, cDNA libraries).

## Results

An overview of the REDI method is presented in Fig 1. First, a DNA library is amplified and transformed into a *MATα* haploid yeast "recipient strain". The genomic integration site consists of a

---

1  Stanford Genome Technology Center, Stanford University, Palo Alto, CA, USA
2  Department of Genetics, Stanford University School of Medicine, Stanford, CA, USA
3  Department of Biochemistry, Stanford University School of Medicine, Stanford, CA, USA
4  Department of Surgery, Harvard Medical School and Massachusetts General Hospital, Boston, MA, USA
5  Laufer Center for Physical and Quantitative Biology, Stony Brook University, Stony Brook, NY, USA
6  Department of Biochemistry and Cellular Biology, Stony Brook University, Stony Brook, NY, USA
7  European Molecular Biology Laboratory (EMBL), Genome Biology Unit, Heidelberg, Germany
   *Corresponding author. Tel: +1 650 721 2976; E-mail: bstonge@stanford.edu
   †These authors contributed equally to this work

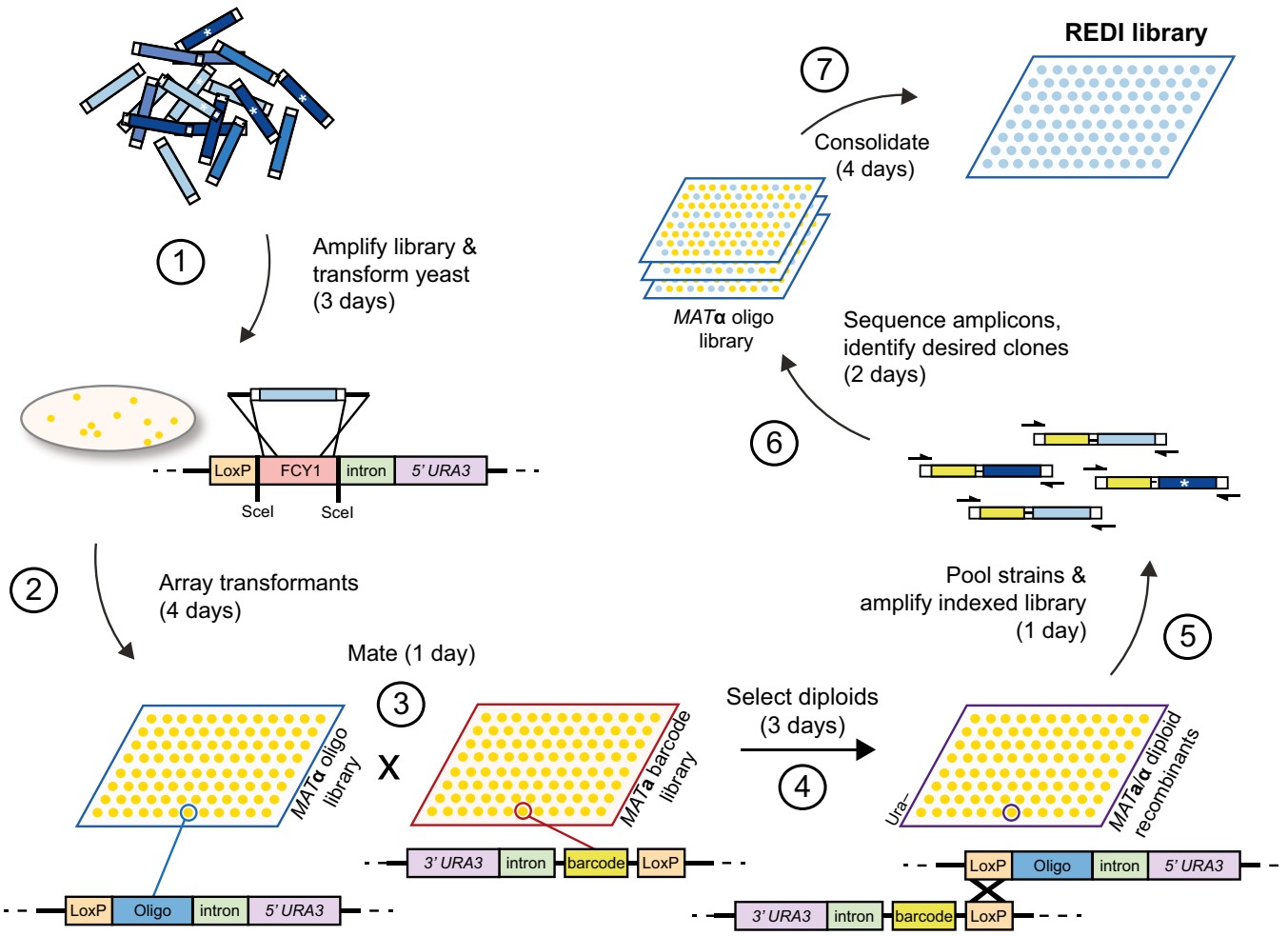

**Figure 1.  Recombinase Directed Indexing (REDI) for high-quality DNA libraries.**
(1) A complex library (e.g. array-synthesized oligonucleotide DNA) is amplified by PCR and integrated into the yeast genome by transformation and homologous recombination. (2) *MAT*α transformants are arrayed in 1,536 format and (3) mated to 1,536 unique *MAT*a barcoder strains. (4) Site-specific (Cre-lox) recombination in diploids physically links the barcode to the exogenous oligonucleotide DNA. (5) Diploid recombinants are combined, and the barcode oligonucleotide locus is amplified by PCR using common priming sites. (6) Amplicons are subjected to paired-end Illumina sequencing. As the barcode assigned to each *MAT*a barcoder strain is known, the plate position of *MAT*α transformants containing oligonucleotide DNA of interest can be readily identified from the sequencing results. (7) Clones are selected and applied in downstream applications of interest. The approximate time required for each step (including yeast growth time) is indicated in parentheses, and is based on processing ~10,000 transformants and isolating ~5,000 clones of interest. Steps (2) and (7) would require additional time if more clones are processed. * indicates DNA synthesis errors.

counter-selectable marker [*FCY1*, which confers sensitivity to 5-fluorocytosine, 5-FC (Ear & Michnick, 2009)], flanked by two *Sce*I restriction sites. Transient expression of the *Sce*I meganuclease during transformation results in DNA double-strand break formation, high-efficiency replacement of *FCY1* by the transforming DNA, and growth on media containing 5-FC (Materials and Methods). The integration site is genetically linked to a partially crippled *loxP* recombination site (*lox71*; Zhang & Lutz, 2002), as well as an artificial intron and the 5′ end of the *URA3* selectable marker (Albert *et al*, 1995; Lee *et al*, 2008; Levy *et al*, 2015). Recipient strain transformants are robotically arrayed and mated to 1,536 uniquely indexed strains (Table EV1). These barcoder strains are *MAT*a haploids that each contain a unique 26-bp barcode identifier genetically linked to a partially crippled *loxP* site (*lox66*; Zhang & Lutz, 2002), an artificial intron, and the 3′ end of the *URA3* selectable marker. These elements are at the same genomic locus as the

integration site in the *MAT*α recipient. Expression of Cre recombinase in the resulting diploids induces recombination between *loxP* sites, which in turn results in physical linkage of the barcode and exogenous library DNA. This event is selectable on media lacking uracil, as a functional *URA3* gene product is reconstituted by intron splicing of the transcribed locus (Levy *et al*, 2015). Single-pot PCR amplification of the indexed library DNA is achieved with common primers, and paired-end Illumina sequencing is used to identify the exogenous DNA linked to each barcode. Employing uniquely indexed sequencing primers allows multiple sets of 1,536 colonies to be interrogated in a single run. As the position of each barcoder strain on the ordered array is known, the sequence data identify the position of recipient strains containing library DNA of interest. Then, a new library can be created by selecting (i.e. "cherry-picking") the desired strains and leaving out those containing undesired sequences, including synthesis errors. DNA of interest can be

isolated through high-fidelity PCR amplification with common primers (Table EV1). Importantly, by combining equal numbers of cells representing each DNA oligonucleotide, REDI has the capacity to generate equimolar libraries. An additional advantage is that clones can be archived for future use, accessed individually, or combined as desired, allowing for customizable sublibraries to be created on demand. Finally, REDI facilitates rapid expression of library members for direct cell-based interrogation of biological function, which is valuable for certain applications. To our knowledge, REDI is the only high-throughput method offering each of these advantages.

As a proof-of-concept, we applied REDI to a library of 7,051 DNA oligonucleotide probes for detection of 354 bacteria related to potable water quality (Table EV1). We designed multiple unique molecular probes (typically 20) for each bacterium, essentially as previously described (Xu *et al*, 2014). Each contained 60 nucleotides (nt) of homology to the target genome, an 8-nt "random" barcode, and 37 nt of internal priming sequence for amplification of probes that circularize upon successful hybridization to their targets (Fig 2A). In addition to these sequence features, each probe was flanked by 20 bases of common priming sequence (for amplification of the library following array-based synthesis) and type IIS

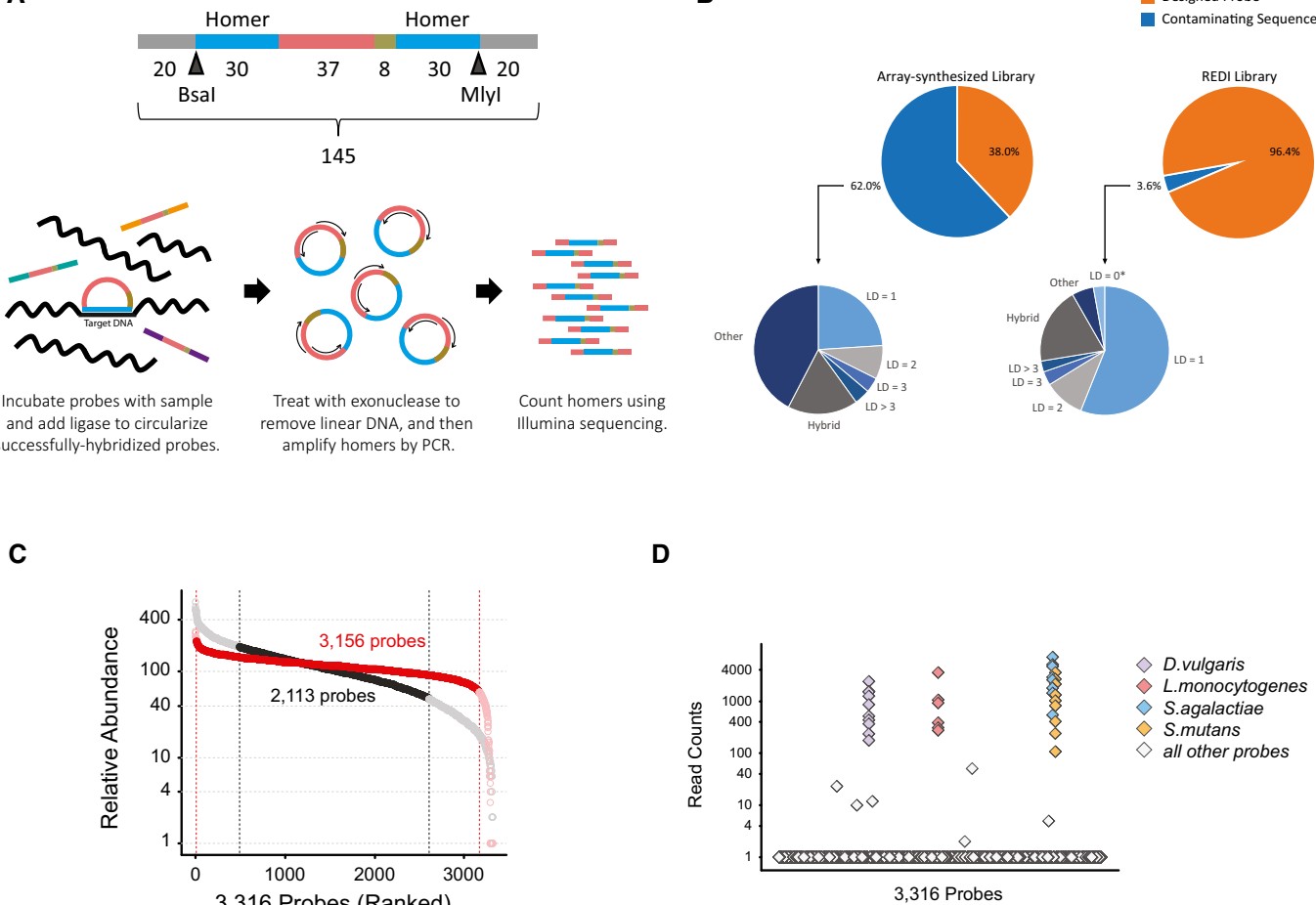

**Figure 2.  Isolation of ~3,300 sequence-verified DNA probes.**

A   Schematic of a molecular probe for detection of bacteria (above). Key features include 20-nt common external priming sites (gray), 2 × 30 nt of homology to the target organism genome ("Homer", blue), internal priming sites for amplification of successfully hybridized probes (red), and a random barcode (green). A probe hybridizing to target DNA is shown below. Hybridized probes are circularized, amplified by PCR, and then quantified using next-generation sequencing.

B   Composition of probe libraries following amplification from array-synthesized DNA (left) or from yeast clones following REDI (right). The percentage of DNA sequences matching a designed DNA probe is shown in orange and those not matching (i.e. contaminating sequence) in blue. Smaller pie charts represent the makeup of contaminating sequences (LD, Levenshtein distance from designed probe; LD = 0*, designed probes not targeted for cherry-picking; Hybrid, a sequence containing homers from two different probes).

C   Relative abundance (*y*-axis) of 3,316 sequence-perfect probes following amplification from array-synthesized DNA (black) or from yeast clones following REDI (red). In both cases, probes are ordered by relative abundance on the *x*-axis. Dotted lines demarcate probes within twofold relative abundance of the median.

D   Multiplex detection of bacterial genomes with 3,316 sequence-verified single-stranded DNA probes. Number of "Homer" reads (*y*-axis) following hybridization of the probe library to a mixture of *Desulfovibrio vulgaris*, *Listeria monocytogenes*, *Streptococcus agalactiae*, and *Streptococcus mutans* genomic DNA. Probes are arranged alphabetically on the *x*-axis, and colored according to their target (see legend).

restriction sites (for removal of common priming sequences and generation of single-stranded DNA probes; Materials and Methods).

We obtained the probe library through commercial, array-based synthesis (CustomArray; Maurer *et al*, 2006), amplified it by PCR, and analyzed its composition using paired-end Illumina sequencing (Materials and Methods; Table EV2). Reported synthesis error rates of 1 in 200 predict that roughly 39% of oligonucleotides in this library (disregarding the common ends and random barcode, and calculated as $1–0.995^{97}$) would contain erroneous probe sequences. Our analysis revealed that ~62% of the amplified library did not match a designed probe (Fig 2B; left), likely reflecting a combination of both synthesis and PCR-derived errors. Many contaminating sequences differed from a designed probe by only 1–3 bases; however, many were also much shorter (i.e. ≤ 95 nt in length) than the designed probes. The latter accounted for > 88% of "Other" sequences. Moreover, and consistent with previous reports (Klein *et al*, 2015), the relative abundance of the designed probes in the amplified library varied widely (Table EV2).

To improve its quality, we applied REDI to this library as outlined in Fig 1. Sequence analysis of ~30,000 indexed transformants identified the plate and position information of 3,316 yeast strains, each containing a unique, sequence-verified molecular probe (oligonucleotide) that had integrated at the target locus. Processing additional transformants was not necessary, as at least one probe for 353 (of a possible 354) bacteria was among the REDI set of 3,316. The exception was *Propionibacterium propionicum*, which was only represented by a single probe in the original designed library. We note that retrieving sequences that are poorly represented in the original transforming DNA (i.e. the array-synthesized library) requires screening a large number of transformants, which adds cost and time to the protocol. For example, based on the composition of the array-synthesized library, we estimate that screening an additional 30,000 transformants would only yield an additional ~1,000 unique probes (Fig EV1).

To evaluate the REDI molecular probe library, we selected colonies from arrayed haploid transformants, consolidated them onto a single agar plate in 6,144-format, and collectively amplified the molecular probes from yeast genomic DNA (Materials and Methods). Illumina sequencing analysis as above revealed a significant improvement in library quality compared to the original array-synthesized library (Table EV3). Specifically, > 96% of the PCR-amplified library contained sequences that perfectly matched one of the 3,316 selected probes (Fig 2B; right). The few sequences that did not match are discussed below. The representation of sequence-verified probes in the new library was also far more uniform (Fig 2C). The REDI library exhibited ~2.2-fold reduction in dispersion (as measured by coefficient of variation) when compared to the same probes amplified from the commercial oligonucleotide pool. In addition, over 95% of individual probes were within twofold relative abundance of the median (Fig 2C). In comparison, only 64% of probes met this criterion in the original array library. We converted the new probe set to single-stranded DNA (Materials and Methods), which was used successfully to detect a mixture of genomic DNA from four bacteria (Fig 2D; Table EV4). Probes directed against other bacteria that amassed a large number of sequencing reads represent false positives (six such probes are evident in Fig 2D). There were also four probes that were false negatives and failed to produce signal, possibly due to design errors

(Table EV4). All of these probes could easily be omitted in future reactions, an advantage of the REDI method over employing array-synthesized probes directly. Similarly, probes directed against bacteria found to be highly abundant in a sample can be removed from the library, enabling detection of organisms present at lower abundance (Xu *et al*, 2014).

Despite the considerable improvements described above, we sought to understand the imperfections in the REDI library by examining the 3.6% of DNA that did not match the expected probe sequences. While sequencing artifacts are expected to account for some of the observed discrepancies, calculated MiSeq error frequencies suggest these effects are negligible (Materials and Methods). Notably, we found that over half of the erroneous DNA differed with a designed probe by only one nucleotide (Fig 2B; right). In principle, erroneous sequences can be synthesis errors that somehow survived the REDI protocol, or polymerase-induced errors generated during PCR amplification of the REDI library. We found evidence for both possibilities. For the latter, single nucleotide errors were likely introduced during PCR, even though a high-fidelity polymerase was used for amplification (Materials and Methods). In addition, PCR-dependent generation of chimeric probes almost certainly contributed to the totality of contaminating DNA, as increasing the number of PCR cycles used for amplification resulted in an increased number of hybrid probes in the library (Fig EV2). Hybrid probes are likely derived from PCR-dependent recombination (Judo *et al*, 1998) at the 37-bp internal amplification site which is common to every probe. Their abundance could probably be further mitigated by stopping the PCR during the exponential phase of amplification.

There was also evidence of oligonucleotide synthesis errors persisting in our REDI library. For example, 49 probes (1.5% of the 3,316 targeted for selection) were absent or poorly represented in the REDI library (< 50 reads each, Table EV3). To analyze these oligonucleotides further, we isolated the 49 colonies representing these probes and then collectively amplified and sequenced their inserts. The results confirmed that most of the desired probe sequences were present, but also revealed that these colonies contained many of the contaminating sequences found in the REDI library (Fig EV3A; Table EV5). Reanalysis of the 49 diploid recombinants found these same sequences, suggesting that these colonies were not clonal (i.e. they were "mixed" colonies; Fig EV3B). Importantly, these data do not support incorporation of multiple oligos in the same cell, a conclusion that we further confirmed by re-streaking a "mixed" colony, and sequencing the inserts from six isolated clones (Fig EV3C). We note that more robust analysis of the diploid recombinant sequences could have flagged these as problematic and would have prevented their inclusion in our library (Materials and Methods). Refining automated colony picking criteria to prevent the picking of colonies that are in contact with one another would also improve matters (Materials and Methods). These, and other strategies for avoiding "mixed" colonies, will reduce contaminants and improve uniformity in future libraries.

To demonstrate the utility of REDI-processed DNA for gene synthesis, we applied the method to isolate and then assemble 14 fragments of the mCherry gene (Fig EV4; Table EV1). Implementing REDI to obtain sequence-verified DNA fragments at near-equimolar concentrations could improve the efficiency of various gene assembly techniques, as well as preclude the need for enzymatic error correction (Kosuri & Church, 2014). While many

applications of REDI do not require recovery of all fragments from the original library, for applications such as gene synthesis that do, uneven and/or poor representation of even a small number of sequences presents a challenge (Fig EV1). More uniform array-synthesized libraries and/or improved strategies for amplifying rare entities will increase the number of sequences recovered from the original library and increase the utility of REDI for these applications.

The relevance of array-synthesized oligonucleotides has increased substantially with the development of the CRISPR/Cas system as a versatile research tool (Doudna & Charpentier, 2014). CRISPR interference (CRISPRi), for example, facilitates programmable gene repression by co-expression of catalytically dead *Streptococcus pyogenes* Cas9 (dCas9) fused to a repressor domain, with a short guide RNA (gRNA) containing 20 nt of complementary sequence to the target (Gilbert *et al*, 2013; Qi *et al*, 2013). Several groups have created effective plasmid-based gRNA libraries for CRISPRi from oligonucleotide pools (Gilbert *et al*, 2014; Smith *et al*, 2016), but as noted above, such libraries suffer from uneven representation, incorrect sequences, and perhaps most importantly, little or no access to individual clones. We therefore sought to create a CRISPRi strain collection for the essential yeast genome using REDI. Various strain collections have proven useful for studying the function of essential yeast genes, but are incomplete (Mnaimneh *et al*, 2004; Breslow *et al*, 2008; Li *et al*, 2011).

We first built a CRISPRi recipient strain containing an integrated dCas9-Mxi1 repressor (Gilbert *et al*, 2013) and a tetracycline-regulatable repressor (TetR) that controls a modified Pol III promoter (*TetO*-P$_{RPR1}$) adjacent to a gRNA library integration site (Smith *et al*, 2016; Fig 3A). Insertion of short oligonucleotides encoding the gRNA target complementarity region produce loci from which fully functional gRNAs are expressed in the presence of the inducing agent anhydrotetracycline (ATc; Fig 3A). Informed by our previous findings for effective gRNA design in yeast (Smith *et al*, 2016), we designed > 18,000 gRNAs to transcriptionally repress 1,117 essential genes (Giaever *et al*, 2002) and 514 genes required for robust respiratory growth (Materials and Methods; Table EV1; Steinmetz *et al*, 2002; Schlecht *et al*, 2014). Following REDI, sequence analysis of ~58,300 tagged transformants identified 9,059 strains containing a unique sequence-verified gRNA. These strains included at least one repressor strain for > 99% (i.e. 1,616 of 1,631) of the genes we targeted, with the vast majority of genes represented by multiple strains (each expressing a unique gRNA; Fig EV5A). These strains were selected from the arrayed transformants, consolidated onto agar plates, archived as individuals, and combined to create pools for competitive growth assays. The majority of strains were present at near-equal concentration following growth in the absence of ATc (Fig EV5B; Table EV6), consistent with tight control of the gRNA promoter by TetR and underscoring the uniformity achieved by creating pools from individually arrayed strains. In addition, when compared to our previous work with plasmid-based CRISPRi libraries (Smith *et al*, 2016), a higher fraction of reads matched a designed gRNA (Fig EV5C and Table EV7), underscoring more efficient use of next-generation sequencing to phenotype strains following competitive growth.

We used competitive assays to measure growth defects resulting from dCas9-mediated gene repression under both fermentative and respiratory growth conditions (Materials and Methods). Upon induction of gRNA expression with ATc, many strains became depleted from the pool. In general, gRNAs directed against genes required for respiration produced growth defects only under respiratory conditions (Fig 3B; Table EV6). Notably, strains expressing gRNAs directed against essential genes also tended to exhibit greater ATc-induced growth defects under respiratory conditions compared to fermentation. This could reflect greater dosage sensitivity or alternatively, more effective gene repression during respiratory growth, which was ~2× slower than fermentative growth. In total, 3,832 gRNAs (43% of those assayed) targeting 1,357 genes (84% of those assayed) induced a growth defect under these conditions (> twofold decrease). It is noteworthy that a substantial number of gRNAs may repress a second gene if that gene's TSS is in close proximity to the intended target, highlighting an important specificity limitation of employing CRISPRi in the open reading frame (ORF)-dense yeast genome (Fig EV5D; Table EV6).

An attribute of REDI is that it provides a direct means for functional interrogation of oligonucleotides in cell-based assays. Using growth as a proxy for transcript levels (assuming that effective repression of an essential gene will negatively impact growth), we examined factors that may influence gRNA efficacy. Consistent with our previous findings (Smith *et al*, 2016), chromatin accessibility and position relative to the transcription start site (TSS) were two important determinants of whether a gRNA exerted effective repression (Fig EV6, Table EV6). The expanded gRNA set refined our previous findings, defining a particularly effective target region between the TSS and roughly 125 nt upstream of the TSS. We also observed a significant correlation between RNA secondary structure and gRNA efficacy, suggesting that interactions between the guide target sequence, constant gRNA sequence, and the leader RNA sequence interfere with target DNA recognition (Fig EV6, Table EV6). These observations can be employed to improve the design of future gRNA libraries, or to select only highly effective gRNAs from the current library for subsequent screens.

Direct access to individual strains is an additional advantage of REDI over employing oligonucleotide libraries directly for CRISPRi screens. For example, the CRISPRi screen identified *IRA1* and *IRA2* as genes whose repression leads to improved growth, even though deletion of these genes was previously shown to result in fitness defects (Giaever *et al*, 2002; Steinmetz *et al*, 2002; Schlecht *et al*, 2014; Fig 3B). Products of the yeast whole-genome duplication, *IRA1* and *IRA2,* are both negative regulators of the RAS/cAMP pathway and orthologs of the human tumor suppressor, neurofibromin 1 (NF1; Tanaka *et al*, 1989, 1990; Ballester *et al*, 1990). Our results extend the previous identification of adaptive mutations in these genes under carbon-limited conditions (Kao & Sherlock, 2008; Wenger *et al*, 2011; van Leeuwen *et al*, 2016) by further showing that perturbing expression also confers a selective advantage. Consistent with the results of the competitive assay, growth curve analysis of individual strains isolated directly from the archived collection revealed accelerated growth of these strains during log phase (Fig 3C). *IRA* gene repression also resulted in early entry into stationary phase (between hours 30 and 45), and a brief exit from stationary phase (between hours 45 and 55). The latter is also observed in control strains, but is accelerated and more pronounced when either *IRA1* or *IRA2* is repressed (Figs 3C and EV7, and Table EV8). These results underscore the utility of REDI libraries for large-scale, highly sensitive, quantitative phenotypic screens, and

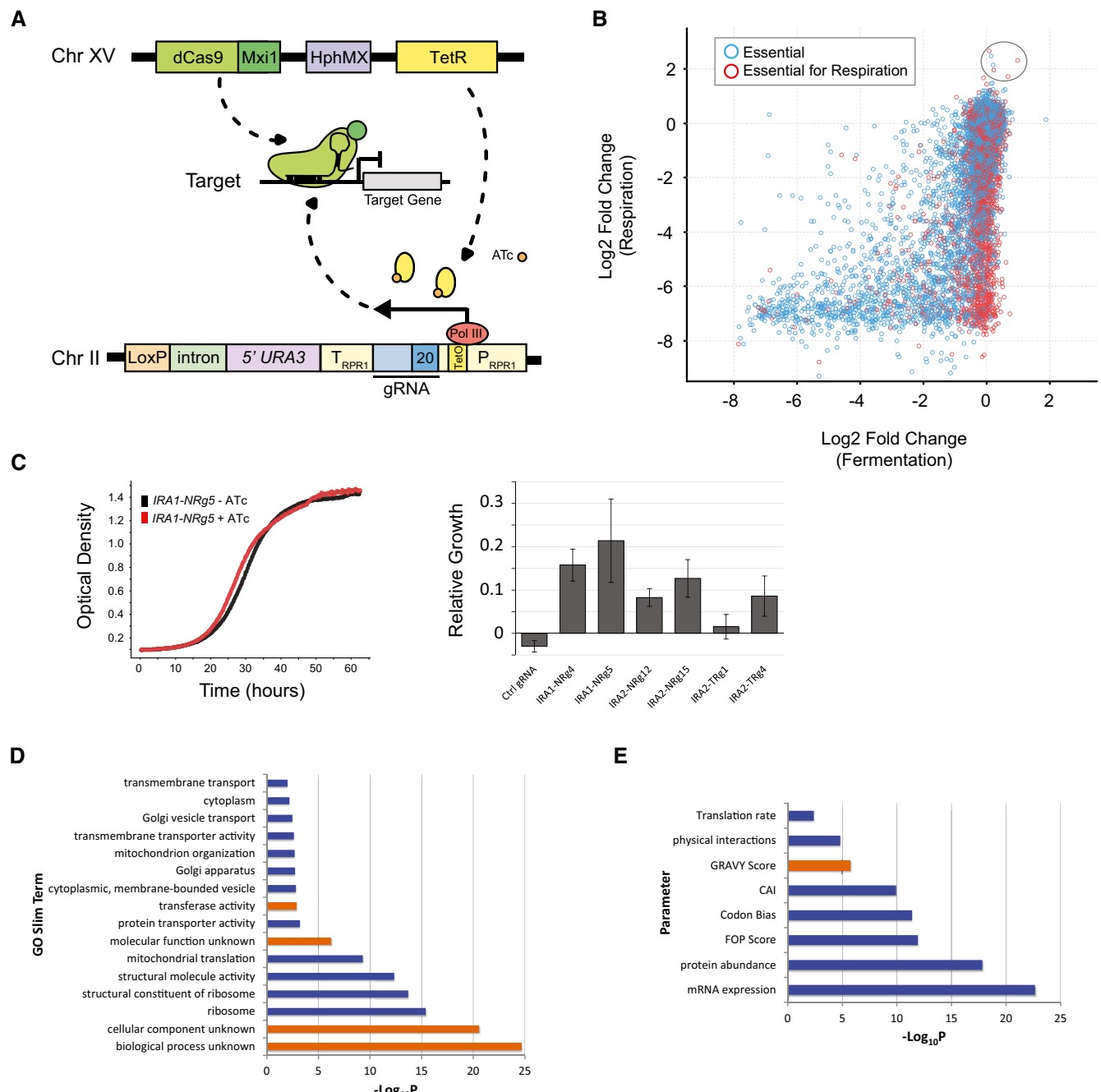

**Figure 3. A CRISPRi strain collection for essential yeast genes.**

A   Cartoon illustrating system for conditional CRISPRi-mediated repression in yeast. The 20-nt region of target complementarity in the gRNA (indicated by "20" in blue) is unique in each strain. In the absence of ATc, TetR binds to the TetO site in the RPR1 promoter to repress Pol III transcription, while the addition of ATc results in gRNA expression.

B   ATc-induced log2 fold change of 8,769 strains (see Materials and Methods) grown in respiratory conditions (YPEG) on the *y*-axis, plotted against ATc-induced log2 fold change of strains grown in fermentation (YPD) conditions. Strains containing gRNAs directed against essential genes are plotted in blue, and those expressing gRNAs directed against genes essential for robust respiration are plotted in red. Six *IRA1* and *IRA2* repressor strains are circled.

C   Representative growth curves for an *IRA1* repressor strain (IRA1-NRg-5) in the presence (red) or absence (black) of ATc are shown on the left. Optical density (OD$_{600}$) is plotted on the *y*-axis over time on the *x*-axis. On the right, growth relative to the no ATc control is plotted for six repressor strains and a control strain. Average of three biological replicates is plotted, and error bars represent the standard deviation (Materials and Methods).

D   Gene Ontology (GO) slim terms exhibiting a significant relationship with CRISPRi-induced sensitivity. Terms plotted in blue are associated with genes whose repression leads to greater growth defects, and those in orange with genes whose repression leads to weaker growth defects. The negative log10-transformed Bonferroni-corrected *P*-value (Kruskal–Wallis test) is plotted on the *x*-axis.

E   As in (D), only a variety of gene characteristics significantly correlated (Spearman rank) with CRISPRi-induced sensitivity are plotted. Parameters in blue are positively correlated and those in orange are negatively correlated.

the advantage of access to individual strains for validating and further exploring screen results.

Additionally, we sought to determine factors that correlated with a gene's likelihood of exhibiting a phenotype when repressed using CRISPRi. We examined both gene ontology (GO) enrichment and a range of biological parameters that might influence sensitivity to transcriptional repression. We found that poorly annotated genes were less likely to show growth defects when repressed (Fig 3D; Table EV9). In contrast, *ribosome*, *structural components of the ribosome*, *structural molecular activity*, and *mitochondrial translation* were each associated with greater sensitivity to CRISPRi-mediated repression. Enrichment analysis of genome-scale CRISPRi results in human cells previously identified *RNP/ribosomal biogenesis* and *translation* (Gilbert *et al*, 2014), suggesting that sensitivity to transcriptional repression of these functional categories is conserved. We also found that highly expressed genes, as defined by a variety of metrics for mRNA and protein abundance, tended to result in greater fitness defects when repressed (Fig 3E; Table EV9).

## Discussion

All steps in the REDI protocol are technically simple and/or amenable to automation. In the current methodology, the most labor-intensive steps are transformation of yeast with the amplified library and preparing samples for Illumina sequencing. Arraying, growing, and replica plating yeast clones are less laborious (as they utilize robotic systems, or simply involve growth in an incubator), but contribute most to the duration of the procedure (typically ~2–3 weeks; Fig 1). As such, the method is perhaps most useful for large-scale projects and not those requiring a small number of DNA clones. The CRISPRi collection (Fig 3), from design through validation, took only a few months to complete and likely required substantially less effort to create than other, similar collections (Mnaimneh *et al*, 2004; Breslow *et al*, 2008; Li *et al*, 2011). REDI reagent costs are minimal, primarily consisting of the array-synthesized oligonucleotide pool, growth medium and plastics for culturing yeast, and sequencing reagents. The last two costs are directly correlated with the number of transformants that must be processed to yield the desired library, which is in turn determined by the composition of the array-synthesized oligonucleotide pool. For the REDI molecular probe library (Fig 2), we estimate that total reagent costs were ~$3,500 (Materials and Methods). While direct cost comparisons are difficult without an accurate accounting of labor and equipment, we note that purchasing 3,316 size-purified (but not sequence-verified) 145-mers (for use as probes, or any other application) would currently cost $384,656 from a leading commercial provider (IDT DNA Ultramer Oligos, https://www.idtdna.com/pages/products/dna-rna/ultramer-oligos).

REDI is not the only cost-effective method that leverages high-throughput sequencing to isolate sequence-perfect DNA from array-synthesized oligo pools (Kim *et al*, 2012; Schwartz *et al*, 2012; Lee *et al*, 2015). For example, dial-out PCR is a method in which random tags are added to oligos and used to identify and then isolate sequence-perfect molecules via high-throughput sequencing and PCR, respectively (Schwartz *et al*, 2012). Unlike REDI, dial-out PCR does not require robotics for yeast colony manipulation and is generally less time-consuming. In addition, dial-out PCR may be superior at isolating rare oligos, as it is not as strongly affected by the initial distribution of oligos in the pool. On the other hand, an advantage of REDI over dial-out PCR is that oligonucleotides can be amplified with the same set of common primers, allowing for facile creation of large or small sublibraries of interest. In dial-out PCR, the number of unique primer pairs (the dominant reagent cost of this method), as well as the number of individual PCR required, directly scale with the number of sequences one wishes to isolate. An additional advantage of REDI is that it is an *in vivo* method and thus supports direct cell-based functional characterization of DNA and the rapid creation of arrayed strain collections. Ultimately, the choice of whether to use REDI or dial-out PCR is dependent on the characteristics of a particular library and the desired downstream applications.

Whether REDI should be employed when working with complex libraries depends on the specific requirements of the application. Array-synthesized oligo pools have been successfully used as molecular probes (Turner *et al*, 2009; Xu *et al*, 2014), to assemble genes (Kosuri *et al*, 2010), and to directly construct CRISPRi plasmid libraries for functional screens (Gilbert *et al*, 2014; Horlbeck *et al*, 2016; Smith *et al*, 2016). Thus, sequence errors and relative abundance variations do not necessarily prevent the acquisition of useful data with these applications. While more efficient utilization of next-generation sequencing (Fig EV5C), eliminating jackpotting and reads wasted on incorrect clones, and other improvements arising from pool uniformity have benefits, they may not always warrant purifying the library with REDI. For certain applications, however, improvements afforded by REDI may be critical. For example, access to individual strains will greatly increase the utility of our CRISPRi collection, allowing individual genes or functional classes of genes to be studied in assays beyond those compatible with the pooled format (Peters *et al*, 2016). The merits of this technology may also prove impactful for high-throughput strain engineering using CRISPR/Cas9 (Doudna & Charpentier, 2014). In principle, oligonucleotide pools encoding guide RNAs and donor DNAs (for homology-directed repair) could be used to introduce precise genetic modifications at scales previously thought to be implausible. REDI could be used in this context to facilitate sequence validation of donor DNA (which would greatly improve editing fidelity), as well as strain isolation, which would greatly expand the possibilities for high-resolution functional analysis.

We describe a technology that offers an extremely cost-effective, high-throughput, rapid method to parse DNA libraries and create arrayed strain collections. REDI has the capacity to produce near-equimolar DNA libraries whose sequence quality is limited only by the fidelity of the polymerase used to amplify the final product. Steps are both scalable and amenable to automation, and because reagent costs are minimal, it can produce high-quality DNA at a fraction of the cost of current column-based synthesis platforms. In addition to parsing array-synthesized oligonucleotide pools, REDI may be further enabled by long-read sequencing technologies (Koren & Phillippy, 2015; Zheng *et al*, 2016) to allow processing of longer DNA fragments such as complex oligonucleotide assemblies, and gene variant or cDNA libraries.

## Materials and Methods

Strains and oligonucleotides used in this study are listed in Table EV1, with the exception of primers used for preparing

Illumina sequencing libraries, which are found in Table EV10. All yeast strains are available upon request.

## Yeast media and reagents

YPD (Yeast extract, Peptone, Dextrose) media consisted of 10 g/l yeast extract, 20 g/l bacto-peptone, and 20 g/l dextrose. Arrayed haploid transformants were generally maintained on YPD containing 20 g/l agar and 200 µg/ml G418. YPEG (Yeast extract, Peptone, Ethanol, Glycerol) medium for competitive growth assays was prepared as follows: 10 g of succinic acid, 10 g of yeast extract, 20 g of bacto-peptone, and 20 g of glycerol were dissolved in 950 ml of distilled water. The pH was adjusted to 5.5 by addition of KOH (potassium hydroxide) pellets. After autoclaving, the medium was allowed to cool to < 60°C, and 25 ml of 95–100% ethanol was added. Agar plates containing 5-fluorocytosine (5-FC) for selection of *fcy1*Δ haploid transformants were prepared by dissolving 43.7 g DOBA (Dropout Agar Base, MP Biomedicals) in 950 ml of water, adding 10 ml of 2 g/l L-methionine (Sigma-Aldrich M-5308), 10 ml of 2 g/l L-histidine (Sigma-Aldrich H-6034), 10 ml of 10 g/l L-leucine (Sigma-Aldrich L-8912), 10 ml of 2.5 g/l L-lysine (Sigma-Aldrich L-5501), and 10 ml of 2 g/l uracil (Sigma-Aldrich U-1128) and then autoclaving. After allowing to cool to 50°C, 2 ml of 50 mM 5-FC (Sigma-Aldrich U-7129) was added for a final concentration of 100 µM 5-FC.

PCR was performed using Q5 high-fidelity DNA polymerase (New England Biolabs), which has a measured error rate of $\sim 1 \times 10^{-6}$ (https://www.neb.com/tools-and-resources/feature-articles/polymerase-fidelity-what-is-it-and-what-does-it-mean-for-your-pcr).

## DNA libraries

The array-synthesized oligonucleotide libraries used in this study are listed in Table EV1 and were purchased as two separate oligonucleotide pools from CustomArray (both using the 12,472 synthesis scale). Molecular probe oligonucleotides and mCherry fragments were derived from a pool in which the DNA concentration was 38.63 ng/µl. CRISPRi guides were derived from a pool in which the total DNA concentration was 53.34 ng/µl. CRISPRi gRNAs were designed to anneal within a window of 0- to 200-nt upstream of the major transcription start site (TSS) of genes (defined as the most common TSS as determined by transcription isoform profiling; Pelechano *et al*, 2013). We designed gRNAs to 1,117 genes that are essential for growth on dextrose (Giaever *et al*, 2002) and 514 genes required for respiratory growth (Steinmetz *et al*, 2002; Schlecht *et al*, 2014). We also designed gRNAs against the non-essential *ADE2* gene. For genes without a defined TSS or with fewer than five gRNAs in the 0- to 200-nt window of TSS, a window from 0- to 300-nt upstream of the start codon was used for gRNA design. Each array-synthesized oligonucleotide contained two separate gRNAs separated by a priming site in the middle that allowed amplification of 10 distinct subpools. gRNAs were organized into subpools based on the number of gRNAs per gene. Then, more effort was spent on those genes for which there were a smaller number of gRNAs per gene. These subpools were first selectively amplified by using the pool-specific primers. A second round of PCR was then performed to extend the overlaps to permit efficient homologous recombination in yeast. Molecular probes were designed essentially as

described previously (Xu *et al*, 2014), with addition of "random" barcodes. The molecular probe library was amplified from yeast genomic DNA using the following PCR conditions: [98°C for 2 min, (98°C for 10 s, 60°C for 40 s, 72°C for 30 s) × 22 cycles, 72°C for 7 min]. The mCherry fragments were designed as described previously (Kosuri *et al*, 2010).

## Barcoder strains

To construct barcoder strains, we generated *MAT*α and *MAT*a "barcode acceptor" strains by inserting two genetic constructs into strains BY4741 (*MAT*a, *his3*Δ1, *leu2*Δ0, *met15*Δ0, *ura3*Δ0) and BY4742 (*MAT*α, *his3*Δ1, *leu2*Δ0, *lys2*Δ0, *ura3*Δ0) via homologous recombination. We replaced the dubious open reading frame *YBR209W* with GalCre-natMX4, where GalCre is a galactose-inducible viral Cre recombinase (Austin *et al*, 1981; Sternberg & Hamilton, 1981), and natMX4 is the dominant nourseothricin resistance marker (Goldstein & McCusker, 1999). The sense strand of the inserted GalCre-natMX4 construct was placed in opposite orientations relative to the centromere in BY4741 and BY4742 to simplify downstream cloning. Deletion of *YBR209W* has been found to have no impact on fitness (Kao & Sherlock, 2008). For each strain, we replaced the counter-selectable *CAN1* gene with the *MFa1*pr-*HIS3*-*MF*α*1*pr-*LEU2* marker (Tong *et al*, 2001; Pan *et al*, 2004; Lindstrom & Gottschling, 2009). The promoters *MFa1*pr and *MF*α*1*pr are only active in *MAT*a and *MAT*α haploids, respectively. Populations of *CAN1/can1::MFa*pr1-*HIS3::MF*α*1*pr-*LEU2* diploids can be easily converted to either *MAT*a or *MAT*α haploids by selecting sporulated cultures on media containing canavanine (for selection against diploids) but lacking histidine or leucine, respectively. Final barcode acceptor strains are SHA345 (*MAT*a, *his3*Δ1, *leu2*Δ0, *met15*Δ0, *ura3*Δ0, ybr209w::GalCre-natMX4, *can1::MFa*pr1-*HIS3*-*MF*α*1*pr-*LEU2*) and SHA349 (*MAT*α, *his3*Δ1, *leu2*Δ0, *lys2*Δ0, *ura3*Δ0, ybr209w:: natMX4-GalCre, *can1:: MFa*pr1-*HIS3*-*MF*α*1*pr-*LEU2*).

We constructed two plasmid libraries that each contain ~100,000 random barcodes. A random barcode is a 26-mer sequence of nucleotides consisting of four random 5-mers (~$10^{12}$ possible variations) interrupted by three constant dimers. The constant dimers allow us to avoid inadvertently creating 6-mer restriction sites for enzymes that are used during plasmid construction. Random barcodes were ordered as oligonucleotides (IDT) and inserted into plasmid backbones by DNA ligation. One plasmid library (U3Kan66) contains a partially crippled *loxP* site (*lox66*; Albert *et al*, 1995; Zhang & Lutz, 2002), the barcode region, the 3′ end of the *URA3* preceded by part of an artificial intron (Lee *et al*, 2008), and the kanMX4-dominant drug-resistant marker (Goldstein & McCusker, 1999). The other plasmid library (U5Kan71) contains a complementary partially crippled *loxP* site (*lox71*; Zhang & Lutz, 2002), the barcode region, the 5′ end of the *URA3* followed by part of an artificial intron (Lee *et al*, 2008), and the kanMX4-dominant drug-resistant marker (Goldstein & McCusker, 1999). We used DNA from these plasmid libraries to replace, by homologous recombination, the natMX4 cassette in SHA345 or SHA349 with *lox66*-Barcode-3′ *URA3*-kanMX4 and *lox71*-Barcode-5′ *URA3*- kanMX4, respectively, to yield SHA345 + BC (*MAT*a, *his3*Δ1, *leu2*Δ0, *met15*Δ0, *ura3*Δ0, ybr209w::GalCre-*lox66*-Barcode-3′ *URA3*-kanMX4, *can1::MF*a1pr-*HIS3*-*MF*α*1*pr-*LEU2*) and SHA349 + BC (*MAT*α,

*his3*Δ1, *leu2*Δ0, *lys2*Δ0, *ura3*Δ0, ybr209w:: kanMX4-5′ URA3-Barcode-lox71-GalCre, can1:: *MFa*1pr-*HIS3*-*MF*α1pr-*LEU2*).

Replica plating yeast in high-density ordered arrays is a robust approach for high-throughput mating (Costanzo *et al*, 2010). Approximately 1,536 SHA345+BC *MAT*a barcoder strains were sequence-validated for this study. Mating of these strains to transformed recipient strains in a 1,536 format, followed by selection for *loxP* recombinants on complete supplement mixture (CSM)-uracil+galactose medium, results in the tagging (i.e. barcoding) of the exogenous DNA incorporated in recipient strains (recipient strains are described below). The genotype of each SHA345+BC strain was verified by assaying for growth on YPD+G418 (for kanMX4), YPD+nourseothricin (for natMX4), and CSM-uracil+galactose following mating to a "tester strain". The barcode sequences of roughly 1,100 of the SHA345+BC strains were identified by Sanger sequencing. The barcode sequences of all SHA345+BC strains were verified/identified using additional barcoder strains of the opposite mating type. Briefly, *MAT*α SHA349+BC strains were mated to SHA345+BC strains in array format, and diploids containing "double-barcodes" were selected on CSM-uracil+galactose medium. Collective amplification of double-barcodes, followed by Illumina sequencing, was used to verify and/or identify the barcode in each SHA345+BC strain.

### Construction of recipient strains

Recipient strains are designed for high-efficiency incorporation of DNA libraries by transformation and the subsequent tagging of these libraries by mating to barcoder strains (described above). All recipient strains used in this study are derivatives of strain #2797, a SHA349+BC strain containing the 26-bp barcode TGCCTAAGCAG GAAGTGTGTTGCAAC. To create a recipient strain, #2797 was modified as follows. First, the *FCY1* gene was replaced with the hphMX4 cassette (Hygromycin B resistance cassette); the hphMX4 cassette was PCR-amplified with primers 957 and 958 (yielding a 1,656-bp product), and transformed into #2797. Transformants were selected on YPD+HygB. HygB-resistant clones were confirmed to grow on Yeast Nitrogen Base (YNB)+Ammonium Sulfate (AS)+Dextrose (Dex)+leucine+histidine+uracil+5-fluorocytosine and were confirmed not to grow on YNB+AS+Dex+leucine+histidine+cytosine (further confirming loss of *FCY1*). The fcy1::hphMX4 deletion was also confirmed by PCR with primers 640 and 641 (which yields a 2,081-bp product). The resulting strain was #2836. Next, the *Sce*I-*FCY1*prom-*FCY1*-*Sce*I cassette was inserted between the *lox71* site and the 26-bp barcode of strain #2836; the 1,089-bp cassette was amplified by PCR with primers 784 and 785 (each contains one *Sce*I site), using the plasmid pJH143 as template, and transformed into #2836. Transformants were selected on CSM-uracil+cytosine+HygB. Successful transformants were confirmed to not grow in YPD+5-fluorocytosine (5-FC), indicating the presence of the *FCY1* cassette, and by PCR with primers P45 and P40 (which yields a 1,414-bp product). The resulting strain, #2849, served as the initial recipient strain and was used in the experiments described in Fig 2.

### "Mitochondria-Repaired" recipient strain

We noted that strain #2849 was prone to spontaneous loss of mitochondrial DNA. This loss, can negatively, and unpredictably, affect growth of transformants. We therefore repaired alleles at three loci known to impact mitochondrial genome stability: *SAL1*, *CAT5*, and *MIP1*. First, we corrected the *sal1-1* allele to wild-type *SAL1* using the mega 50:50 method (Horecka & Davis, 2014) using primers SAL1.80.1 and SAL1.80.2 together with PCR template pJH140 (Table EV1). A correct recombinant was identified by genomic PCR and confirmed by Sanger sequencing. The resulting strain is JHY627. Next, we converted *CAT5*(91I) to *CAT5*(91M) using the mega 50:50 method and primers CAT5.80.1 and CAT5.80.2 together with PCR template pJH140. A correct recombinant was identified by genomic PCR and confirmed by Sanger sequencing. The resulting strain is JHY629. Finally, we converted *MIP1*(661A) to *MIP1*(661T) using a two-step allele replacement strategy using pLND44-4, as described by Dimitrov *et al* (2009). A correct recombinant was identified by Sanger sequencing of both the *MIP1* quantitative trait locus (QTL) and a region 988-bp downstream that has a known plasmid error. The resulting strain is JHY650. DNA from the three QTLs was PCR-amplified from the final strain, JHY650, and Sanger-sequenced to confirm the desired alleles. All alleles were correctly repaired/replaced, and the pLND44-4 plasmid sequence error noted by Dimitrov *et al* (2009) was not present. In addition, the barcoding/artificial-intron region of the JHY650 strain was confirmed by Sanger sequencing.

### CRIPSRi recipient strain

A recipient strain for guide RNAs targeting yeast genes for transcriptional repression by dCas9-Mxi1 was created. This strain facilitates expression of fully functional gRNAs from a tetracycline-regulatable *RPR1* promoter (Bak *et al*, 2010), and constitutively expresses a dCas9-Mxi1 fusion protein. To create this strain, we first created a cassette containing *Sce*I-*FCY1*prom-FCY1-*Sce*I flanked by the *RPR1* promoter and the structural (i.e. common) guide region and *RPR1* terminator. The sequence of this integration cassette is available at: https://benchling.com/s/qpwEfqmX/edit. Briefly, *Sce*I-*FCY1*prom-FCY1-*Sce*I DNA was amplified by PCR with 128-pRPR1-FCY1-fwd and 129-gRNA-FCY1-rev from genomic DNA isolated from the original recipient strain (#2849). The cassette was inserted into the *Not*I restriction site of pRS416gT-Mxi1 plasmid (Smith *et al*, 2016; Table EV1). From this plasmid, a 1,769-bp cassette was PCR-amplified with primers 131-pRPR1-int-fwd and 130-RPR1t-int-rev and transformed into strain #2836. The cassette was inserted by homologous recombination into a region adjacent to the *URA3* marker, thereby replacing the kanMX4 marker. Transformants were selected on YNB+AS+Dex+leucine+histidine+cytosine. Transformants were screened for loss of G418 resistance due to loss of the kanMX4 cassette, and successful integration was confirmed by PCR using primers 894 and 895 (which yielded a 2,183-bp product). The resulting strain was #2869. Next, the hphMX4 cassette at the *FCY1* locus was replaced with kanMX4 by PCR amplification of kanMX4 (with primers 142 and 143 (1,282 bp), transformation into strain #2869, and selection on YPD medium containing G418. This resulted in strain #2877.

A cassette was created to enable the incorporation of dCas9-Mxi1 into strain #2877. First, the hphMX4 cassette was amplified by PCR using the primers 133-Tef1-HphMX-fwd and 132-HphMX-rev, and then inserted in pRS416gT-Mxi1 (Smith *et al*, 2016) between dCas9 and the Tet repressor with Gibson assembly. The sequence and map

of this plasmid are available at: https://benchling.com/s/2Gki8et5/ edit. Primers 134-Mid-HphMX-fwd and 137-site18dn-M13F-rev were used to amplify the dCas9 portion of the plasmid, and primers 135-Mid-HphMx-rev and 136-site18up-GPM1-fwd were used to amplify the TetR portion of the plasmid. Each PCR product contained overlapping fragments of the hphMX4 cassette. These two PCR products were co-transformed into strain #2877, and integrated into site 18 (YORWdelta17 XV) from Flagfeldt *et al* (2009). This site had been characterized as an appropriate site to express heterologous proteins. Transformants were selected overnight in YPD+HygB liquid medium and further selected on YPD+HygB agar plates. Successful integration of the DNA was confirmed by PCR with primers 138-Site18-YORWdelta17-fwd and 139-Site18-YORW-delta17-rev, which anneal outside the site and amplify only if the proper product is present. This strain was named yACJ1 (Table EV1). Effective CRISPRi in this strain was confirmed by transforming it with gRNAs directed against the *SEC14* and *ERG25* genes, whose repression is known to produce growth defects (Smith *et al*, 2016).

We improved yACJ1 as described above with #2849. We used the 50:50 method (Horecka & Davis, 2014) to alter alleles at two of the loci known to impact mitochondrial genome stability. We repaired the *sal1-1* allele to wild type using primers SAL1.80.1 and SAL1.80.5 together with PCR template pJH136. Integration of the 50:50 cassette was selected for by growth on CSM-URA media and confirmed by PCR with primers SAL1.80.3 and URA3.34.6. Loss of the *URA3* marker in 5-fluoroorotic acid (FOA)-resistant colonies was determined by PCR from genomic DNA with primers SAL1.80.3 and SAL1.80.4, followed by Sanger sequence confirmation of the wild-type allele. We used the *SAL1*-corrected strain as the starting strain to repair the *CAT5*(91I) allele to *CAT5*(91M) in the same manner, using CAT5.80.1 and CAT5.80.5 primers for 50:50 cassette construction, and CAT5.80.3 and URA3.34.6 primers for confirmation. FOA-resistant colonies were Sanger-sequenced using CAT5.80.3 and CAT5.80.4 primers to identify the *CAT5*(91M) isolates. These two allele changes naturally present in the wine isolate *RM11*, and reduce the frequency of petite formation by eightfold (Dimitrov *et al*, 2009).

### Sequence analysis of diploid recombinants

Paired-end Illumina sequencing was used to identify the exogenous DNA physically linked to each barcode in the diploid recombinants. For each plate of recombinants, total genomic DNA was purified from ~150 to 200 million cells using the YeaStar Genomic DNA Kit (Zymo Research) and following the manufacturer's directions. The genomic region of interest was PCR-amplified from 1 μl of purified genomic DNA template [98°C for 2 min, (98°C for 10 s, 60°C for 20 s, 72°C for 30 s) × 30 cycles, 72°C for 7 min], producing a 429-bp amplicon (molecular probe library) or a 941-bp amplicon (CRISPRi library). Typically, 240 cycles were used to sequence the exogenous DNA in the forward direction, and 60 sequencing cycles were used to sequence the barcode in the reverse direction. Briefly, sequencing reads were binned according to short (i.e. six nucleotide) indexing barcodes included in both the forward and reverse sequencing primers (index pairs were unique to each plate of diploid recombinants), and, then, by the 26-bp barcode (unique to each barcoder strain and, thus, colony position in the arrayed plate

of recombinants). These steps required sequences to perfectly match the designed barcodes (all other sequences were excluded from further analysis). The exogenous DNA sequence in each colony was then identified as the most commonly observed sequence, between the common priming regions, in each set of binned reads. This sequence was compared to the designed molecular probe or guide RNA libraries, and sequences perfectly matching a designed sequence were prioritized for selection. We further refined this list by de-prioritizing clones where the perfect match sequence was supported by fewer than 50 reads. In some cases, we also filtered out diploid recombinants having multiple different sequences that perfectly matched multiple different designed sequences (likely indicating robotic picking of a "mixed colony"). Extending this last filter to include partially matching sequences will further mitigate contamination from mixed colonies (Fig EV3) and will improve library quality.

### Sequence analysis of DNA library composition

Paired-end Illumina DNA sequencing was used to examine the composition of DNA libraries produced during this study. Amplification of the molecular probe library with DirMP-fwd and DirMP-rev primers produced amplicons of 276 bp in length. These amplicons were paired-end-sequenced (2 × 250 cycles) to cover the entire length of the molecular probe sequences in both directions. To mitigate the confounding effect of sequencing-derived errors, analyses were restricted to sequences located between the molecular probe common amplification sites (TAGACGTAAGCCTGGTCTCA and ATCGGGAATCGAGTCTACCT) where the forward and reverse reads were in perfect agreement. In addition, the same sequencing run was used to analyze and compare the array- and yeast-derived libraries. The calculated sequencing error rates for this run (based on PhiX control DNA that was included in the sample) were 1.78 and 2.63% for Read 1 and Read 2, respectively. Based on these rates, sequencing artifacts are predicted to contribute ~0.01% of the observed errors (0.0178 × 0.0263 × 0.25).

Notably, an 8-nt random barcode was included in the design and synthesis of the molecular probes. These sequences become clonal after transformation into yeast and are no longer useful for improving quantification of target DNA (as they were originally intended). As the sequences of these barcodes are not known beforehand, they were excluded from the library composition analysis.

Sequencing analysis of the molecular probe libraries revealed that 61.8% of the array-synthesized library and 3.6% of the library produced by REDI did not match any of the probes that were targeted for selection (Fig 2B). We refer to these DNA sequences as "contaminating DNA". These sequences were analyzed for similarity to a designed probe using bowtie2 (version 2.2.8) and samtools (version 1.2), and for hybrids (i.e. sequences where the first Homer maps to one probe, and the second Homer maps to a different probe).

For the library produced by REDI, we excluded sequencing reads that were associated with 66 clones that were selected (i.e. cherry-picked) but later determined to have a "petite" phenotype (reduced growth due to spontaneous loss of mitochondrial DNA). We attempted to remove these clones by replica plating the entire collection onto growth medium that only supports respiration prior to creating the probe library. We still observed low levels of sequencing reads from these clones in the library. As they do not represent

true contaminating DNA, they were excluded from the analysis presented in Fig 2.

To make a robust comparison of the uniformity between the two libraries, we compared the 3,316 probes from the REDI library to the same 3,316 probes from array-synthesized library. Using R (version 2.15.2), we also down-sampled the number of reads used to analyze the REDI library to match that of the CustomArray library, and computed the Coefficient of Variance (CoV, computed as the standard deviation divided by the mean) using the down-sampled data. We repeated the same process 100 times to get the mean and 95% confidence interval of the CoV of the REDI library (mean: 0.317, 95% CI: 0.314–0.321). The CoV of the array-synthesized library was 0.709.

### Estimating recovery rate of sequence-perfect DNA from array-synthesized libraries

We used the *in silico* random sampling (*sample* function in R (version 2.15.2); replace = FALSE) to estimate how many perfect molecular probes could be retrieved by processing increasing numbers of transformants. Specifically, we randomly sampled DNA sequences found in the amplified CustomArray library, and counted how many sequence-perfect probes were recovered in different sample sizes. This simulation used the DNA library composition (as determined by "*Sequence analysis of DNA library composition*" described above) following amplification with PCR parameters that were similar to those used for producing the DNA employed in the actual transformation. As the transforming DNA was also size-selected prior to transformation, DNA fragments > 130 bp or < 80 bp were excluded from the sampling to better mimic the transformation conditions.

### Sequence analysis of probes missing or poorly represented from the molecular probe library

We identified 49 molecular probes that were absent or poorly represented in the molecular probe library following parsing in yeast. These probes were identified by fewer than 50 sequencing reads in the DNA library composition analyses. To explore explanations for the poor representation of these probes, we selected the 49 colonies representing these probes, collectively amplified the exogenous DNA (i.e. the amplified array-synthesized oligonucleotides that integrated into the target locus of the recipient strain), and sequenced those amplicons using MiSeq (Illumina). Sequences matching those found in the composition analyses of the original library were identified and are presented in different colors in Fig EV3A. An examination of diploid recombinant sequences associated with the 49 indexing barcode pair/barcode combinations of the poorly represented clones are given in Fig EV3B. Sequences represented by fewer than five reads were excluded from these analyses.

### Selection of sequence-verified clones from arrayed haploids

All strains were selected (i.e. cherry-picked) from the arrayed collection of haploid transformants using the Stinger (Singer Instruments), and consolidated onto agar plates. For the molecular probe library, each plate contained ~300 clones plus a control strain (which did not contain the integration locus present in our recipient strains) at border positions (i.e. the first and last columns and rows on the

plate). As colonies at these positions tend to grow faster (Costanzo *et al*, 2010), the border strain was intended to reduce the potential for introduction of abundance biases among clones containing a molecular probe. Following 1 day of growth at 30°C, plates were replicated onto a single agar plate in 6,144 format. The CRISPRi library was similarly created, but without a border strain. The 14 mCherry clones were selected onto agar in 96-position plate format. All selected colonies were archived at −80°C in YPD + 15% glycerol.

### Competitive growth assays

Yeast culturing and sample collection were performed using a cell-screening platform that integrates temperature-controlled absorbance plate readers, plate coolers, and a liquid handling robot. Pipetting events were triggered automatically by Pegasus Software and performed by a Freedom EVO workstation (Tecan). The 9,059 strains of the CRISPRi collection were screened in two separate experiments; pool1 containing 8,337 strains and pool2 containing 722 strains (Table EV6). Each condition was assayed in three biological replicates. Briefly, 700 μl of yeast cultures was grown under fermentative (YPD) or respiratory (YPEG) conditions, in the presence or absence of 250 ng/ml ATc, in 48-well plates at 30°C with orbital shaking in Infinite plate readers (Tecan). To maintain small-volume culture growth over many doublings, 80 μl of the culture was removed when it reached an optical density (OD) of 0.76, added to a well containing 620 μl of sterile medium, and then allowed to grow further. After three such dilutions, 600 μl of the culture was collected and saved to a 4°C cooling station (Torrey Pines) when it reached an OD of 0.76. This procedure amounted to ~10 culture doublings from the beginning of the experiment. Yeast genomic DNA was purified using the YeaStar Genomic DNA Kit (Zymo Research) from the collected cells and used as a template for PCR with indexed (i.e. barcoded) sequencing primers flanking the gRNA sequence. PCR products were confirmed to be the correct size by agarose gel electrophoresis, and, then, combined and cleaned with magnetic beads (Thermo Scientific). Sequencing was performed with an Illumina MiSeq, using either 300 or 500 v.2 cycle kits.

### Analysis of competitive growth assay data

Counts were generated by enumerating all sequences with perfect matches in both Read 1 and Read 2 of paired-end MiSeq data (i.e. requiring that the first 60 nucleotides of Read 1 containing the barcode and gRNA target complementarity were also present without mismatch in Read 2 (reverse complement)). These sequences were then mapped to the gRNAs that we had designed. Given the nature of our experimental design, we expect many strains to exhibit growth defects upon addition of ATc, and, consequently, produce a major change in the distribution of the resulting pool composition. This phenomenon created a unique challenge to the data normalization that is required when comparing across separate experiments. Widely used normalization methods, such as quantile normalization that assumes an unchanged shape of the distributions, or z-score normalization that assumes common mean (or median) across samples, cannot be applied.

To address this challenge, we applied a normalization strategy that relies on the identification of a set of "neutral" guides. Based

on the experimental design, we make the following two assumptions: (i) many of the guides are damaging, i.e. lead to reduced fitness of the cells, and hence a reduced proportion of the guides in the pool; (ii) a smaller but significant portion of the guides are neutral, i.e. they do not change the fitness of the cells, but their relative proportion will slightly increase in the pool due to the decreased fitness of the other strains. Based on these two assumptions, the following three steps were taken to define neutral guides:

1  We normalized all samples by the relative values of their respective total sequencing read counts.

2  We clustered the log-transformed data using Euclidean distance measure and complete linkage algorithm, and chose a distance cutoff value of 11 to obtain the major clusters.

3  We calculated the average log counts of all guides in each cluster and selected the cluster with the highest average counts as the neutral cluster. Guides in the neutral cluster had slightly increased counts in the ATc samples when compared to the non-ATc samples, consistent with the expected behavior of neutral guides.

Finally, data normalization is performed on the counts using the neutral guides as reference, i.e. $X_{gs} \leftarrow \frac{X_{gs}}{S \cdot X_{Rs}/\sum_{s=1}^{S} X_{Rs}}$, where $X_{Rs}$ is the average counts of the neutral guides in the neutral cluster $R$ identified above, $s$ indicates the samples, $g$ indicates individual guides, and $S$ is the total number of samples.

Raw sequencing counts, normalized counts, and ATc-induced $\log_2$ fold change values (calculated from the normalized counts averaged across biological replicates) were calculated separately from two sets of competitive growth experiments, one involving pool1 ($n = 8,337$ strains) and one involving pool2 ($n = 722$ strains). Results were merged afterward. For the analyses presented in Fig 3, we excluded gRNAs that had a sum of < 75 counts between the three no ATc replicates ($n = 237$ for YPD; $n = 254$ for YPEG), as well as gRNAs targeted to dubious ORFs ($n = 28$) and the non-essential *ADE2* gene ($n = 6$). All gRNAs were annotated with nucleosome occupancy, gRNA midpoint, normalized chromatin accessibility (ATAC-seq), gRNA location, gRNA midpoint distance, and nearby TSS data determined using a webtool (http://lp2.github.io/yeast-crispri/; Smith *et al*, 2016). The gRNA midpoint distance generated by the webtool includes both experimentally determined TSS positions calculated from transcription isoform profiling data (Pelechano *et al*, 2013) and approximate TSS positions for which experimental data did not exist. Guide positioning analysis (Fig EV6) only uses data from experimentally determined TSSs and excludes all gRNAs that target regions within 150 bp of the major TSS of two or more genes. In the cases where a single gRNA could be targeting two distinct genes in our target set, the gRNA was annotated (i.e. named) based on the gene whose TSS was closest to its midpoint. The RNA structure free energies in kcal/mol were generated with RNA fold (version 2.2) in the Vienna RNA Package 2.0 (Lorenz *et al*, 2011) and included the 85-nt RNA leader from the *RPR1* promoter "gtccctatcagtgatagagatggcgcacatggtacgctgtggtgctcgcg gctgggaacgaaactctgggagctgcgattggcag", the variable 20-nucleotide targeting sequence, and the sequence of the constant part of the gRNA "gttttagagctagaaatagcaagttaaaataaggctagtccgttatcaacttgaaaaag tggcaccgagtcggtgctttttt". The annotated datasets with counts and $\log_2$ fold change values for each gRNA are available in Table EV6. Figures were generated using python packages matplotlib, scipy,

and numpy. Statistical analyses were performed with scipy.stats package using the spearman r function. Figure EV6 was generated using matplotlib hist2d, bins = 40, norm = LogNorm(). Additionally, in Fig EV6, for the plot of TSS distances, analysis was constrained to only include data for gRNAs that fall between 0- and 300-nt upstream of the TSS. This excluded 108 gRNAs. For the analysis of gRNAs potentially influencing multiple ORFs, gRNAs were considered to have a potential second target if the TSS of another ORF was within 150-bp (upstream or downstream) of the gRNA midpoint, a conservative estimate based on our findings indicating an ideal gRNA targeting window between the TSS and 125-nt upstream of the TSS.

To identify factors that influence a gene's susceptibility to producing a growth phenotype when repressed, we first calculated the average CRISPRi-induced log2 fold change of each gene by averaging the log2 fold change values of all guides targeting that gene. We compared these values across Gene Ontology (GO) Slim terms for process, component, and function obtained from the Saccharomyces Genome Database (SGD; Cherry *et al*, 2012) using a Kruskal–Wallis test. Additionally, we compared these average log2 fold change values against a variety of biological parameters using Spearman rank correlation. These parameters included RNA-seq expression levels in YPEG (Jiang *et al*, 2014), protein abundance data obtained from PaxDb (Wang *et al*, 2012), the number of physical and genetic interactions determined by counting the number of unique interacting partners listed on SGD, as well as a variety of other parameters listed on SGD including the codon adaptive index (CAI), codon bias, Frequency of Optimal Codons (FOP), hydropathicity of protein (GRAVY score), aliphatic index, instability index, isoelectric point (pI), protein length, and molecular weight. Translation rate was calculated from ribosomal profiling data for each gene (Artieri & Fraser, 2014). Bonferroni-corrected *P*-values were calculated by dividing the initial *P*-value by the number of tests performed.

We compared the percentage of reads mapping to designed gRNAs from all competitive growth experiments using the broad tiling plasmid pool (Smith *et al*, 2016) sequenced in the "pRS416gT-mxi1-H1-H27″ MiSeq run, to all competitive growth experiments using our pooled REDI CRISPRi collection sequenced in the "NorcadiaCP-R2877H-RH-mcherry-Ess6″ MiSeq run. Analyses were restricted to those sequence reads that had perfect sample indices, perfect constant sequences flanking the 20 nt of gRNA sequence, and where the first 60 nt of Read 1 was contained exactly within Read 2 as defined above. We then totaled the counts for all sequences and determined the fraction that were perfect matches (using python) to gRNAs from the Broad Tiling set and the REDI CRISPRi collection, respectively. The data are available in Table EV7.

### Pure strain growth assays and analysis

Isogenic growth assays were conducted in 96-well microtiter plates. Optical density was measured every 15 min over the course of > 60 h using an Infinite or GENios microplate reader (Tecan). The growth rate of a strain was calculated as follows: (i) the first OD reading was subtracted from all OD readings of the corresponding curve to set the baseline of the growth curve to zero, and (ii) the area under the curve (AUC) was then calculated as the sum of all (background-subtracted) OD readings. In all cases, average AUC

was calculated from six technical replicates (the same strain grown under the same condition in the same plate). "Relative growth" (used in Figs 3C and EV7) was calculated essentially as previously described (Schlecht *et al*, 2012) as follows: $(AUC_{ATc} - AUC_{DMSO})/AUC_{DMSO}$, where $AUC_{DMSO}$ represents growth in the absence of ATc that was assayed on the same microtiter plate as the ATc-treated culture.

### Molecular probe preparation and reactions

Molecular probes were designed as described previously (Xu *et al*, 2014). Following PCR amplification from yeast genomic DNA, the molecular probe library was converted to single-stranded DNA as previously described (Xu *et al*, 2014). Briefly, the double-stranded 145-bp PCR product was digested with *Bsa*I (New England Biolabs) for 2 h at 50°C, to create a 5′ overhang. *Bsa*I was heat-inactivated for 20 min at 65°C. The digested product was treated with shrimp alkaline phosphatase (New England Biolabs) for 1 h at 37°C. The alkaline phosphatase was heat-inactivated for 15 min at 80°C. That product was digested with *Mly*I (New England Biolabs) for 2 h at 37°C. *Mly*I digestion produces a blunt end. *Mly*I was heat-inactivated for 15 min at 65°C. The product of this reaction was treated with Lambda exonuclease (New England Biolabs) for 15 min at 37°C to generate the desired single-stranded 105-mer DNA molecules. Lambda exonuclease was heat-inactivated for 15 min at 80°C. Following enzymatic modification of the DNA, removal of the common priming sites was confirmed using a Bioanalyzer (Agilent). The 5′ ends of the DNA fragments were phosphorylated using polynucleotide kinase (New England Biolabs) for 30 min at 37°C. The polynucleotide kinase was heat-inactivated at 65°C for 15 min.

The molecular probe library was incubated with denatured target DNA under hybridization conditions overnight. Target DNA consisted of $3.79 \times 10^{-11}$ µmol of *Desulfovibrio vulgaris* DNA, $5.36 \times 10^{-11}$ µmol of *Listeria monocytogenes* DNA, $11.4 \times 10^{-11}$ µmol of *Streptococcus agalactiae* DNA, and $8.58 \times 10^{-11}$ µmol of *Streptococcus mutans* DNA. All target DNAs were purchased from the American Type Culture Collection (ATCC). Where sufficient DNA base sequence homology existed between the Homer of the probe and the target DNA, a 60-bp duplex was formed, with the 5′-phosphate of the probe immediately adjacent to the 3′-hydroxyl. One unit of Ampligase was added to the DNA, and the reaction incubated for 10 min at 58°C. The sample was treated with exonucleases I and III for 15 min at 37°C. The exonucleases were heat-inactivated for 15 min at 80°C. Successfully hybridized Homer sequences were PCR-amplified from exonuclease-resistant circular DNA with primers MP-Up3 and MP-Dn1 (Table EV10) and enumerated using Illumina sequencing. Although it was originally designed to improve quantification, the 8-nt "random" barcode was rendered clonal by our method. Thus, it was not used in these analyses.

### Recipient strain transformation

Recipient strains were cultured in 5 ml of YPD overnight to saturation. The next day, the culture was diluted into 20 ml of YPD to a concentration of 0.25 OD/ml and grown for ~4 h at 30°C with moderate agitation until it reached ~1 OD/ml. Cells from 7.25 ml of culture were collected by centrifugation at 1,300 *g* for 2 min in a swinging bucket rotor and washed twice with 1 ml of 100 mM lithium acetate (LiAc). The cell pellet was resuspended in 500 µl of salmon sperm DNA (2 mg/ml); 40 µl of this cell suspension (containing roughly 15 million cells) was transferred to a 1.5-ml Eppendorf tube. To that tube, ~1 pmol of linear transforming DNA and ~1 pmol of linear DNA containing the constitutive *Saccharomyces cerevisiae TEF1* promoter, the *SceI* ORF, and the *S. cerevisiae CYC1* terminator were added. The total volume of the DNA solution added did not exceed 60 µl. Both linear DNAs were produced by PCR. The transforming DNA was generally amplified from 0.5 µl or 1 µl of array-synthesized library (CustomArray) using primers containing homology to the target locus of the recipient strain (Table EV1) and size-purified using the Pippin Prep size selection system (Sage Biosciences) prior to transformation. Tubes were incubated at 30°C for 30 min; 200 µl of a solution containing 45% polyethylene glycol (PEG) and 100 mM LiAc was added and incubated for an additional 30 min at 30°C. Cells were heat-shocked at 42°C for 60 min in a water bath, collected by centrifugation at 14,000 rpm for 1 min and resuspended in 4 ml of YPD. Cells were recovered at 30°C with moderate agitation for 18 h and plated on square agar plates containing 5-FC. This protocol typically produced ~$10^6$ transformants. Plating 1 µl of recovered culture resulted in ~200–300 colonies per plate, which was optimal for our automated colony picking robot. Notably, the recovered culture could be stored at 4°C for several weeks without significant reduction in colony-forming units.

### Automated colony picking

An automated system was used to enable high-throughput re-arraying of yeast transformants. Briefly, square 15 × 100 × 100 mm petri dishes are grasped by the robot's input arm, de-lidded, and moved to an imaging station. The plate is top lit by an annular light, and an image of the plate is captured with a 480 × 480 pixel CCD camera. The image is analyzed by a custom C program. The image is first smoothed and averaged to obtain a background value representative of the agar medium and plastic plate. A threshold based on proportion of the maximum values and the background is used to identify any object edges (i.e. colonies). An additional filter rejects all objects that do not meet size, roundness, brightness, or proximity criteria. The user can set the object rejection/colony acceptance criteria from an image viewing interface by adjusting roundness, size, proximity, and smoothing stringency. Once selected, these values are automatically applied to the rest of the picking session. The center coordinates of colonies are adjusted using a pre-calibrated image distortion algorithm, and these values, transformed into the robot's input arm, are used to drive the arm in the x–y plane. Colonies are picked into 384-well microtiter plates using a rotating head holding 20 stainless steel needles. The needles are rinsed with water and heat-sterilized between picking events. The system picks ~1,000 colonies per hour. Inoculated microtiter plates are then incubated overnight at 30°C and stamped to 1,536 format onto solid medium using a ROTOR (Singer Instruments).

### Reagent cost estimate for REDI molecular probe library

The oligonucleotide pool cost was estimated to be ~$1,300 and was calculated by multiplying the fraction of molecular probe

oligonucleotides in the library (7,051 of 12,472) by the total cost of the library. Yeast media and plastic costs were estimated to be ~$1,000 and were based on processing the ~30,000 transformants that were re-arrayed onto 29 PlusPlates (Singer Instruments). The cost of yeast transformation reagents was also included in this estimate. Illumina sequencing costs were estimated to be ~$1,200 and were calculated by multiplying the fraction of reads used in each of three sequencing runs, by the cost of the MiSeq reagent kits. The costs of additional reagents needed to prepare the sequencing library (yeast genomic DNA extraction, PCR, bead clean-up, quantitative PCR) were also included in this estimate.

### Data availability

Meta data for Illumina DNA sequencing runs are listed in Table EV10. The Illumina sequencing data supporting the results are available in the Gene Expression Omnibus (GEO) repository, GSE81094; http://www.ncbi.nlm.nih.gov/geo/query/acc.cgi?acc = GSE81094.

**Expanded View** for this article is available online.

### Acknowledgements

The authors are grateful to Mia Jaffe for the control strain used in Figs 3 and EV7, to Leo Parts for help with generating Table EV6, to Marie Evangelista for critically reading the manuscript, and to Billy Lau and other members of the Ji Research Group for helpful discussion. This work was supported by grants from the US National Institutes of Health (P01HG000205 to L.M.S. and R.W.D., U01GM110706-02 to R.W.D., and R01HG008354 to S.F.L.) and by the The Louis and Beatrice Laufer Center.

### Author contributions

JDS and RPS wrote the manuscript. US, WX, JH, MJP, RSA, AC, YFL, LMS, and SFL contributed to writing the manuscript. RPS, JDS, and US performed experiments and analyzed data. WX, YFL, and KR analyzed data. RWH, SS, RAOB, JH, AC performed experiments. MJP developed the automated colony picker. JH developed the library transformation protocol. SFL developed the loxP barcoding system. SFL, LMS, and RWD provided essential insight and advice.

### Conflict of interest

The authors declare that they have no conflict of interest.

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
