## [Review Process File · Molecular Systems Biology]

A Method for High-throughput Production of Sequence-verified DNA Libraries and Strain Collections

Justin Smith, Ulrich Schlecht, Weihong Xu, Sundari Suresh, Joe Horecka, Michael Proctor, Raeka Aiyar, Richard Bennett, Angela Chu, Yong Li, Kevin Roy, Ronald Davis, Lars Steinmetz, Richard Hyman, Sasha Levy and Robert St. Onge

Corresponding author: Robert St. Onge, Stanford University

Review timeline:

Submission date:	28 July 2016
Editorial Decision:	14 September 2016
Revision received:	01 December 2016
Editorial Decision:	10 January 2017
Revision received:	12 January 2017
Accepted:	12 January 2017

Editor: Maria Polychronidou

Transaction Report:

1st Editorial Decision

14 September 2016

Thank you again for submitting your work to Molecular Systems Biology. We have now heard back from the three referees who agreed to evaluate your study. As you will see below, the reviewers think that the presented approach seems interesting. However, they raise a number of concerns, which we would ask you to address in a revision of the study.

REFeree REPORTS

Reviewer #1:

Summary

In this study the authors describe an indexing, sequencing and cherry-picking technique, "REDI" to generate arrayed yeast culture collections containing sequence-verified DNA of interest from array-synthesized oligos. While this is demonstrated in yeast with a recombination-based indexing strategy, one could imagine that this general approach could be utilized in any desired laboratory organism (i.e. *E. coli*) with a similar indexing and cherry-picking strategy. The major advantages of this technique are the ability to pool oligos in a more equimolar fashion than directly off array, pool in desired sub-pools, and reduce the number of oligos containing errors (as the authors show, by more than an order of magnitude). The authors demonstrate this technique with three separate applications, first by generating an DNA probe library, second by assembling the mCherry gene, and

third by generating a yeast essential gene CRISPRi knockdown collection. This approach addresses important technological roadblocks in array synthesis technology (pooled nature of sequences, low sequence fidelity, and unequal pooling), and has implications for high throughput DNA library construction and downstream assays.

General remarks

While the approach addresses important issues inherent to array synthesis technology, the major limitation of this manuscript is that the authors do not quantitatively demonstrate the advantages of their technique over the current state of the art. Without this demonstration, the impact of the work is unclear. For example, to what extent would the DNA probe application work directly off the array and what are the major shortcomings in using sequence-imperfect, non-uniform DNA? When assembling the mCherry gene, what is the improvement in the percent of clones that contain the correctly assembled gene from array DNA to sequence verified DNA? For the yeast CRISPRi library, how does a higher percentage of sequence verified clones quantitatively improve the ability to do pooled screens and interrogate biological systems (for example, in comparison with the authors' previous pooled yeast CRISPRi study - Smith et al., *Genome Biology* 2016)? Are there larger-scale applications that can be highlighted with access to individual clones? One could imagine that most shortcomings of sequence imperfect, non-uniform DNA can be overcome with larger sequencing depths. While the advantages of the authors' technique are plausible, a direct functional comparison of REDI libraries to array libraries in at least one application would greatly improve the significance and impact of this work.

The generation of the 9,059 member yeast CRISPRi library is also notable. Other CRISPRi culture collection efforts are significantly smaller (for example <300 members in a *B. subtilis* library - Peters et al., *Cell* 2016). This is a major outcome of this work, and could serve as an important resource in the field.

Major points

- 1) Timing (particularly colony picking/cherry picking steps) could potentially be an important limitation to the described technique. The authors do mention the speed of their colony picking apparatus in the methods section, but speed for the rest of the protocol and considerations of the timeline should be added to the main text in an appropriate section.
- 2) The authors' discussion of the source of errors in their library is appreciated. The contribution of error sources such as polymerase errors and PCR chimeras is generally not well described. The authors should include information from Figure S1 as to the source of the "contaminating" sequences in Figure 2 (perhaps as an expansion of the pie-charts in panel B) as this is discussed extensively in the text and is an important aspect of their analysis. The high percentage of "other" sequences in the original library is also intriguing - the predicted source of these sequences or better characterization of their source should be added as this may be of general interest with the use of array synthesis technologies.
- 3) The discussion (lines 254-264) and data (Figure 3D) pertaining to design rules underlying gRNA is interesting and an important outcome of this work. However, it seems somewhat tangential from the main narrative of the manuscript (describing a novel array-based library generation technique) and is similar to their previous publication (Smith et al., *Genome Biology* 2016). This data should still be included, but perhaps it could be replaced in the main text with biological insights enabled uniquely by the REDI technique.
- 4) Figure 3C (growth curve of IRA1 CRISPRi strain) should show data from overlaid/averaged replicate experiments performed on different days.
- 5) In Figure S4, step 5 (image of mCherry expression in yeast) appropriate negative controls (WT yeast) should be included.

Minor points

- 1) The authors should include more quantitative language in the abstract, replacing vague

descriptors (specifically, 41: 'remarkable uniformity', 43: 'vast majority')

2) The calculation of "<37% of 200-mer DNA being of correct sequence" (line 66) should be justified with appropriate calculation as in the results section (line 125-126) either in the main text or as a supplementary note.

3) Numbering each step of the REDI protocol in Figure 1 could help with the interpretation and parsing of this figure. The barcode symbol used throughout to denote the 1536 unique barcodes is unclear/non-standard and should be replaced or a label should be added, to improve interpretation. Finally, mention of the timeline of the protocol (i.e. day 1, day 2 .. day N) could be added to the figure to better illustrate the protocol.

4) In line 143, the word "unique" should be added in front of "probes" as this would clarify the statement (assuming this is what the authors meant).

5) In line 152, the authors should state the exact fold improvement in CV. Furthermore, given that for most applications one would care about all oligos in a given library, the CV (rather than quartile CV/ robust CV etc.) is almost certainly the appropriate measurement. Additionally, in line 155, the authors should state the percentage of probes within two-fold relative abundance of the median in the original library to directly compare results.

6) Figure 2 could be revised to improve presentation of the data. In panel A, the authors should consider adding clarifying how the circularized probe assay works (as presentation of the probe does not convey how the assay works and is confusing as presented on its own). In panel B, the authors should add information regarding the discussion of contaminating sequences, and further break down the predicted source of contaminating sequences for each library, as discussed in the main text (see major point #2). Panels C and D contain repeated information and should be combined and made consistent across both libraries. For example, the median and two-fold relative abundance lines could be drawn on the ordered distribution plot for both libraries, and percentage of probes within this area could be directly written on the plot. It is essential that characterization of both libraries is displayed in identical formats to facilitate a direct comparison between the two.

7) The authors should describe specific bioinformatics tools used to perform quality filtering, trimming etc. of sequences in the methods section.

Reviewer #2:

Review of "Characterization of clonal DNA", Smith .. St. Onge

This manuscript introduces a method the authors name Recombinase Directed Indexing, or REDI.

The method is motivated by recent technologies for array-based DNA synthesis. This technology can create massive libraries of 100-200 nt sequences, but with drawbacks that synthesis errors lead to incorrect fragments, and simultaneous release of all fragments simultaneously leads to a complex mixture. Often sub-libraries have to be amplified out.

The core of the REDI method is a creative approach that uses yeast mating and loxP recombination to physically link a synthetic DNA sequence with a barcode, and in turn to link the barcode to the physical location of a yeast clone. Other similar ideas have been used powerfully for high-throughput screens, such as yeast two-hybrid physical interaction screens and genetic screens. I say this to point to the power of the method, not to suggest any lack of innovation.

The manuscript describes several successful applications of the method. The first example is generating a systemic library of probes to detect pathogenic bacteria. They sequence about 30K indexed transformants to identify ~3K correct, unique library members from a library designed to have 7K members. The 10x over-sequencing and 50% recovery seems to be a reasonable performance for a first attempt.

The main power of the method is in generating large libraries that don't require 100% completion. A corresponding weakness is that the method isn't useful on its own for applications where every designed sequence must be recovered. Thus, it works well for creating input libraries for Cas/CRISPR but not as well for entry into large-scale synthetic biology gene and pathway synthesis. The authors discuss these points adequately. I suspect that REDI could be used as a first-pass synthesis step for synthetic biology, with a second REDI step or a different method used to fill in the fragments that are not recovered.

Overall, I find the manuscript quite strong and compelling. The technical analysis of the data is very complete. There are some biological findings, but really the strength of the manuscript is the method rather than any new biological understanding. I have only two minor requests.

First minor request, p. 7, line 182: 'mixed' colonies are described as non-clonal colonies. I'd like the authors to discuss this point a little further. Can they determine whether these were double-picks, so the well contains two clones? Or did two synthetic DNA fragments enter a single yeast cell? I suspect it's the double-pick but would like to know more explicitly. Obviously this is easy to test by restreaking the mixed colony or by looking at the DNA sequence.

Second request: Can the authors say anything explicitly about enabling other groups to use this method? For example, are the arrayed, bar-coded yeast strains available from the authors or from another source?

Reviewer #3:

Summary

The authors develop a method to isolate arbitrary pools of sequence-verified array-synthesized oligonucleotides. The authors use this method to produce high quality DNA oligonucleotides for three applications: (1) molecular probes for detecting genomic DNA from bacteria of relevance to potable drinking water, (2) oligos for synthesizing genes, and (3) gRNAs for CRISPR/Cas9 applications (CRISPRi of essential genes in yeast). In addition to validating the "REDI" technology, the authors used their CRISPRi data to analyze parameters that could potentially impact gRNA efficacy.

General remarks

This is not the first technology capable of selecting sequence-verified oligos synthesized on low cost DNA microarrays. While the technology has some nice features (e.g., the ability to produce arbitrary pools of oligonucleotides and the ability to normalize the relative abundance of library members), the authors could do more to explicitly explain why their method is superior to previous methods like Dial-Out.

An advantage this method has over Dial-Out is that it does not require a large primer repository. Since individual clones contain monoclonal oligo sequences, as long as the researcher (or robot) is willing to pick the correct clones, one common set of primers can be used to amplify an arbitrary library composition. This could be useful, especially for applications in requiring empirical feedback and re-pooling. However, this is a double-edged sword because the main drawback of the method seems to be the need to monoclonally manipulate and sequence an enormous number of yeast clones, which requires robotics. This will likely limit the wide-spread use of the method.

It's not clear that the oligos need to be perfect or uniformly represented for the applications described in the manuscript. The authors should provide evidence that perfect probes are necessary for their applications. Otherwise, it would just be easier, faster, and cheaper to prepare libraries from crude microarray DNA.

The authors devote a lot of the text to describe how a tiny fraction of their REDI library had mistakes in it. It's not really clear why such a small number of mistakes would be problematic or

why this discussion occupies so much of the manuscript (perhaps much of it could be moved to the supplement). It would be nice if the authors would provide evidence for why such mistakes are such a liability for their method.

After sequencing 30,000 clones the authors observed only 3,316 unique sequences with perfect clones, which covers only 47% of their desired sequences. As the authors point out, doubling the number of clones sequenced would only give them an additional ~1000 oligos. So does that mean that REDI is only useful for applications that require only 47-61% of the designed oligos?

Major points

It is appreciated that the authors report cost and time required for this method, but given the fact that other similar methods exist it is not adequate to just compare chip-based synthesis vs. individual oligo synthesis (and even so, it seems a little disingenuous to say that all 3,316 oligos would be required to target 354 genomes). Molecular Inversion Probes have been around for a long time, and people have synthesizing them on microarrays for years. What is it about their application that requires higher-quality probes? It would be more appropriate to compare REDI vs. similar existing methods like Dial-Out. They should also discuss how REDI addresses problems that are not already addressed by Dial-Out. While the authors mention three main features of REDI (capacity to generate equimolar libraries, ability to archive strains, and ability to directly deploy library members for cell-based interrogation of biological function), it is not clear that these are unique advantages of REDI. For example, Dial-Out could arguably be superior with regard to archiving, since desiccated DNA can be very cheaply and very stably stored for long durations on filter paper.

As the authors point out in Figure S1, they are likely observing PCR-dependent recombination of their oligos, resulting in "hybrid" oligos. Their method could be improved by doing qPCR of the oligo pools so that they can stop the PCR during exponential amplification (i.e., before the PCR plateaus). As is, it's hard to evaluate how much of a problem DNA synthesis is (they can't control that) vs. DNA amplification (they can control that).

I'm not totally convinced by Figure 2C. The authors do not report which of the original 7,051 probes from the array-synthesized library are observed in the 3,316 REDI probes. It would be useful to color the array-synthesized library probes that are also observed in the REDI library as one color, and the non-represented library probes as another color. Additionally, plotting the array-synthesized and REDI libraries' abundances on the same axes would make a clearer figure (i.e., there is 0 abundance in the REDI library for more than half of the total probes). Anyway, the calculations of uniformity should always be compared for the exact same subset of probes in both libraries (i.e., if you count a probe with an abundance of 10 in the array-synthesized library and it is not present in the REDI library, then you need to count the abundance in the REDI library as 0). It's not totally clear from the manuscript, but at least based on the figure, it doesn't appear that the authors did this.

Figure 2E: Does this data represent any quantitative information about relative population frequencies of the different bacteria? Not only would that be more useful, it would also be a compelling reason that the probes would need to be uniformly represented in the REDI library. Also are the "other probes" with non-zero read counts false positives? Could the authors validate probe specificity empirically, and then remove these probes from the library for future use? Do the authors ever observe probes that don't work (i.e., false negatives)?

Minor points

Line 64: "accumulate" probably isn't the best word. It would be clearer to say that the probability of at least one synthesis error increases with each additional base added.

Figure 1 is confusing as currently laid out, making the method look like an iterative cycle. It would be helpful to spatially separate the first and last steps to avoid this confusion.

Do the authors have any hypotheses for why the essential genes that are not related to respiration still seem to show less depletion in anoxic conditions? Is it an artifact, or a biological insight?

"The CRISPRi collection 285 (Figure 3), from design through validation, took only a few months to complete and 286 required substantially less effort than what would be needed with traditional methods."-This claim needs to be qualified somehow. Some existing methods use microarray oligo pools without the added steps of REDI. It seems like better performance would be a better argument, but then the authors would need to actually compare performance of their REDI library vs. a non-sequence-verified library.

1st Revision - authors' response

01 December 2016

Reviewer Comments

Reviewer #1:

Summary

In this study the authors describe an indexing, sequencing and cherry-picking technique, "REDI" to generate arrayed yeast culture collections containing sequence-verified DNA of interest from array-synthesized oligos. While this is demonstrated in yeast with a recombination-based indexing strategy, one could imagine that this general approach could be utilized in any desired laboratory organism (i.e. *E. coli*) with a similar indexing and cherry-picking strategy. The major advantages of this technique are the ability to pool oligos in a more equimolar fashion than directly off array, pool in desired sub-pools, and reduce the number of oligos containing errors (as the authors show, by more than an order of magnitude). The authors demonstrate this technique with three separate applications, first by generating a DNA probe library, second by assembling the mCherry gene, and third by generating a yeast essential gene CRISPRi knockdown collection. This approach addresses important technological roadblocks in array synthesis technology (pooled nature of sequences, low sequence fidelity, and unequal pooling), and has implications for high throughput DNA library construction and downstream assays.

General remarks

While the approach addresses important issues inherent to array synthesis technology, the major limitation of this manuscript is that the authors do not quantitatively demonstrate the advantages of their technique over the current state of the art. Without this demonstration, the impact of the work is unclear. For example, to what extent would the DNA probe application work directly off the array and what are the major shortcomings in using sequence-imperfect, non-uniform DNA? When assembling the mCherry gene, what is the improvement in the percent of clones that contain the correctly assembled gene from array DNA to sequence verified DNA? For the yeast CRISPRi library, how does a higher percentage of sequence verified clones quantitatively improve the ability to do pooled screens and interrogate biological systems (for example, in comparison with the authors' previous pooled yeast CRISPRi study - Smith et al., *Genome Biology* 2016)? Are there larger-scale applications that can be highlighted with access to individual clones? One could imagine that most shortcomings of sequence imperfect, non-uniform DNA can be overcome with larger sequencing depths. While the advantages of the authors' technique are plausible, a direct functional comparison of REDI libraries to array libraries in at least one application would greatly improve the significance and impact of this work.

We thank the reviewer for making this important point. It is true that sequence-imperfect, non-uniform DNA probe pools constructed from array oligos can detect bacteria (as we have previously shown in Xu *et al.* 2014) and that gene assembly from array oligos is readily achieved using enzymatic error correction. We used these applications in our manuscript primarily as a means to develop the REDI method and to characterize its capabilities. For our final application, we decided to produce a CRISPRi strain collection (vs. pool) because it represents a resource whose construction was uniquely enabled by REDI. As the reviewer points out, we have successfully used pooled screens with plasmid libraries before (Smith et al, *Genome Biology* 2016). We now include Figure EV5C demonstrating more efficient use of NGS data for phenotyping our REDI library compared to our previously published plasmid library. The analyses are described in the Methods (lines 773-782). This shortcoming of the

plasmid library can be overcome with larger sequencing depths, but this increases the overall cost of phenotyping the pool. That said, we believe the most important advantage afforded by REDI in this case is direct access to individual clones in the library – much like the yeast deletion collection, or other similar collections. This allows the exploration of individual genes or functional classes of genes in assays beyond those compatible with the pooled format (e.g., microscopy). Indeed, we have already been approached by multiple researchers to provide subsets of the collection. Moreover, optimized sub-collections can be designed and extracted based on e.g., best-performing guide RNAs; this attribute also applies to molecular probe libraries. In cases where certain microbial species dominate samples, the corresponding probes could be removed to allow detection of lower-abundance species. We believe these features greatly increase the overall value of REDI collections/libraries to the research community. We also point out REDI's potential for future large-scale genome editing, in which oligo errors could result in incorrect edits and off-target effects – and in which access to individual clones would greatly expand opportunities to functionally characterize variants of interest.

We have added a paragraph (lines 343-361) to the discussion which better describes the advantages of REDI in the context of different applications.

The generation of the 9,059 member yeast CRISPRi library is also notable. Other CRISPRi culture collection efforts are significantly smaller (for example <300 members in a *B. subtilis* library - Peters et al., Cell 2016). This is a major outcome of this work, and could serve as an important resource in the field.

We agree and are hopeful that this collection will be a valuable resource. As requested by reviewer #2 below, we have added a statement regarding the distribution of this collection to the community on line 375. We are also working on depositing these strain resources at the ATCC.

Major points

1) Timing (particularly colony picking/cherry picking steps) could potentially be an important limitation to the described technique. The authors do mention the speed of their colony picking apparatus in the methods section, but speed for the rest of the protocol and considerations of the timeline should be added to the main text in an appropriate section.

We agree and have added the timing of each step in Figure 1. We have also added a textual description of timing considerations at lines 310-316.

2) The authors' discussion of the source of errors in their library is appreciated. The contribution of error sources such as polymerase errors and PCR chimeras is generally not well described. The authors should include information from Figure S1 as to the source of the "contaminating" sequences in Figure 2 (perhaps as an expansion of the pie-charts in panel B) as this is discussed extensively in the text and is an important aspect of their analysis. The high percentage of "other" sequences in the original library is also intriguing - the predicted source of these sequences or better characterization of their source should be added as this may be of general interest with the use of array synthesis technologies.

We have expanded the piecharts in Figure 2B to include a breakdown of the contaminating sequences as suggested. The "other" sequences were predominantly truncated oligos (now described on lines 129-133). As a result of this change, we have also eliminated much of the original Figure S1. The data showing the effect of PCR cycles on generating 'hybrid' sequences is now condensed into one plot and presented in Fig.EV2.

3) The discussion (lines 254-264) and data (Figure 3D) pertaining to design rules underlying gRNA is interesting and an important outcome of this work. However, it seems somewhat tangential from the main narrative of the manuscript (describing a novel array-based library generation technique) and is similar to their previous publication (Smith et al., GenomeBiology 2016). This data should still be included, but perhaps it could be replaced in the main text with biological insights enabled uniquely by the REDI technique.

This is a good point, and we have moved this figure to the supplement (now Fig.EV6). As we feel the results will be of interest to readers, we left much of the discussion of these results in the main text, slightly modified for clarity (see lines 261-274).

4) Figure 3C (growth curve of IRA1 CRISPRi strain) should show data from overlaid/averaged replicate experiments performed on different days.

We thank the reviewer for this suggestion. We repeated the assay and now plot the average relative growth (see Methods at lines 783-794) from 3 biological replicates (performed on different days). These new data revealed that the IRA1/2 repressors exhibited reproducible, ATc-dependent accelerated growth in log phase that was not observed in the control. These new data are now included in Figure 3C. Additional data including replicate data for the 'post-saturation hump' is included in Figure EV7. Raw data are included in Table EV8. These results are discussed at lines 275-292.

5) In Figure S4, step 5 (image of mCherry expression in yeast) appropriate negative controls (WT yeast) should be included.

We now include images of untransformed BY4743 (WT) as a negative control to figure EV4.

Minor points

1) The authors should include more quantitative language in the abstract, replacing vague descriptors (specifically, 41: 'remarkable uniformity', 43: 'vast majority')

We have modified the abstract accordingly at lines 40-44.

2) The calculation of "<37% of 200-mer DNA being of correct sequence" (line 66) should be justified with appropriate calculation as in the results section (line 125-126) either in the main text or as a supplementary note.

We have added this calculation in the main text on line 67.

3) Numbering each step of the REDI protocol in Figure 1 could help with the interpretation and parsing of this figure. The barcode symbol used throughout to denote the 1536 unique barcodes is unclear/non-standard and should be replaced or a label should be added, to improve interpretation. Finally, mention of the timeline of the protocol (i.e. day 1, day 2 .. day N) could be added to the figure to better illustrate the protocol.

We thank the reviewer for these suggestions to make Figure 1 easier to understand and have modified it accordingly. Specifically, we have numbered the protocol steps, replaced the barcode symbol, and have added time required to complete each step.

4) In line 143, the word "unique" should be added in front of "probes" as this would clarify the statement (assuming this is what the authors meant).

This is indeed what we meant, and we have made the change.

5) In line 152, the authors should state the exact fold improvement in CV. Furthermore, given that for most applications one would care about all oligos in a given library, the CV (rather than quartile CV/ robust CV etc.) is almost certainly the appropriate measurement. Additionally, in line 155, the authors should state the percentage of probes within two-fold relative abundance of the median in the original library to directly compare results.

Based on the suggestion by reviewer #3 to compare the same probes in each library (see additional discussion below), the fold improvement in CV is now 2.2 fold. We have updated the text and removed references to quartile CV. We also include a comparison of the percentage of probes within two-fold relative abundance of the median in Figure 2C, which was ~64% in the original array-synthesized library. This is described at lines 154-158.

6) Figure 2 could be revised to improve presentation of the data. In panel A, the authors should consider adding clarifying how the circularized probe assay works (as presentation of the probe does not convey how the assay works and is confusing as presented on its own). In panel B, the authors should add information regarding the discussion of contaminating sequences, and further break down the predicted source of contaminating sequences for each library, as discussed in the main text (see major point #2). Panels C and D contain repeated information and should be combined and made consistent across both libraries. For example, the median and two-fold relative abundance lines could be drawn on the ordered distribution plot for both libraries, and percentage of probes within this area could be directly written on the plot. It is essential that characterization of both libraries is displayed in identical formats to facilitate a direct comparison between the two.

We appreciate this reviewer's suggestions to improve Figure 2. Based on this advice, we have made several changes to Figure 2. We have expanded the cartoon in Fig.2A to clarify how the probe assay works. We have added pie charts to Fig.2B to describe the breakdown of contaminating sequences. We have combined the separate plots in Fig.2C into one. The revised Fig.2C plots both the abundance of the 3,316 probes from REDI, and the same 3,316 probes from array synthesized library (as requested by reviewer #3). Those probes within 2x abundance of the median are highlighted for both sets. It should be noted that to ensure a fair comparison, we down-sampled the number of Illumina reads used to characterize the REDI library. This is described in the methods on lines 621-628. We have removed Fig.2D from the manuscript entirely because it was, as the reviewer indicated, redundant with Fig.2C.

7) The authors should describe specific bioinformatics tools used to perform quality filtering, trimming etc. of sequences in the methods section.

We have added more descriptions of the tools used in the methods at lines 610-611, 623, 630-631, 739-745, 749-750, and 780-782.

Reviewer #2:

Review of "Characterization of clonal DNA", Smith .. St. Onge

This manuscript introduces a method the authors name Recombinase Directed Indexing, or REDI.

The method is motivated by recent technologies for array-based DNA synthesis. This technology can create massive libraries of 100-200 nt sequences, but with drawbacks that synthesis errors lead to incorrect fragments, and simultaneous release of all fragments simultaneously leads to a complex mixture. Often sub-libraries have to be amplified out.

The core of the REDI method is a creative approach that uses yeast mating and loxP recombination to physically link a synthetic DNA sequence with a barcode, and in turn to link the barcode to the physical location of a yeast clone. Other similar ideas have been used powerfully for high-throughput screens, such as yeast two-hybrid physical interaction screens and genetic screens. I say this to point to the power of the method, not to suggest any lack of innovation.

The manuscript describes several successful applications of the method. The first example is generating a systemic library of probes to detect pathogenic bacteria. They sequence about 30K indexed transformants to identify ~3K correct, unique library members from a library designed to have 7K members. The 10x over-sequencing and 50% recovery seems to be a reasonable performance for a first attempt.

The main power of the method is in generating large libraries that don't require 100% completion. A corresponding weakness is that the method isn't useful on its own for applications where every designed sequence must be recovered. Thus, it works well for creating input libraries for Cas/CRISPR but not as well for entry into large-scale synthetic biology gene and pathway synthesis. The authors discuss these points adequately. I suspect that REDI could be used as a first-pass synthesis step for synthetic biology, with a second REDI step or a different method used to fill in the fragments that are not recovered.

Overall, I find the manuscript quite strong and compelling. The technical analysis of the data is very complete. There are some biological findings, but really the strength of the manuscript is the method rather than any new biological understanding. I have only two minor requests.

First minor request, p. 7, line 182: 'mixed' colonies are described as non-clonal colonies. I'd like the authors to discuss this point a little further. Can they determine whether these were double-picks, so the well contains two clones? Or did two synthetic DNA fragments enter a single yeast cell? I suspect it's the double-pick but would like to know more explicitly. Obviously this is easy to test by restreaking the mixed colony or by looking at the DNA sequence.

The data are consistent with double-picks (not 2 oligos entering the same cell). As suggested by the reviewer, we demonstrated this by re-streaking a mixed colony, and Sanger-sequencing the DNA inserts from six isolated colonies. These results are now included in Fig.EV3C and described on lines 194-201.

Second request: Can the authors say anything explicitly about enabling other groups to use this method? For example, are the arrayed, bar-coded yeast strains available from the authors or from another source?

We have added a statement (on line 375) that we will distribute all strains upon request. We are currently working to deposit our collections at ATCC, which houses several yeast genome-wide collections, to maximize their utility for the community.

Reviewer #3:

Summary

The authors develop a method to isolate arbitrary pools of sequence-verified array-synthesized oligonucleotides. The authors use this method to produce high quality DNA oligonucleotides for three applications: (1) molecular probes for detecting genomic DNA from bacteria of relevance to potable drinking water, (2) oligos for synthesizing genes, and (3) gRNAs for CRISPR/Cas9 applications (CRISPRi of essential genes in yeast). In addition to validating the "REDI" technology, the authors used their CRISPRi data to analyze parameters that could potentially impact gRNA efficacy.

General remarks

This is not the first technology capable of selecting sequence-verified oligos synthesized on low cost DNA microarrays. While the technology has some nice features (e.g., the ability to produce arbitrary pools of oligonucleotides and the ability to normalize the relative abundance of library members), the authors could do more to explicitly explain why their method is superior to previous methods like Dial-Out.

An advantage this method has over Dial-Out is that it does not require a large primer repository. Since individual clones contain monoclonal oligo sequences, as long as the researcher (or robot) is willing to pick the correct clones, one common set of primers can be used to amplify an arbitrary library composition. This could be useful, especially for applications in requiring empirical feedback and re-pooling. However, this is a double-edged sword because the main drawback of the method seems to be the need to monoclonally manipulate and sequence an enormous number of yeast clones, which requires robotics. This will likely limit the wide-spread use of the method.

We agree that a discussion of the advantages of REDI over similar methods like dial-out PCR was lacking in our manuscript, and have added a paragraph accordingly (lines 326-342).

It's not clear that the oligos need to be perfect or uniformly represented for the applications described in the manuscript. The authors should provide evidence that perfect probes are necessary for their applications. Otherwise, it would just be easier, faster, and cheaper to prepare libraries from crude microarray DNA.

Please see our first response to reviewer #1 and the new paragraph we have added to the discussion (lines 343-361). We have now clarified that the merits of REDI will warrant its use for some applications but not for others. For the CRISPRi library, for example, access to individual strains is probably the greatest advantage of REDI. Without it, our follow-up experiments with the IRA1/2 genes would have been much more difficult. Additionally, a CRISPRi collection offers the advantage that researchers with an interest in a specific subset of genes can readily isolate those strains for their experiments. In fact, we have already shared subsets of strains with several collaborators. Additionally, arrayed collections can be used to study phenotypes that are not readily accessible in pooled growth experiments, such as cell morphology phenotypes. We hope that this and other contextual advantages of REDI are now more clear.

The authors devote a lot of the text to describe how a tiny fraction of their REDI library had mistakes in it. It's not really clear why such a small number of mistakes would be problematic or why this discussion occupies so much of the manuscript (perhaps much of it could be moved to the supplement). It would be nice if the authors would provide evidence for why such mistakes are such a liability for their method.

We included this discussion because the types of mistakes we observe suggest that, with further improvement, the number of errors could be reduced even further. We feel that this discussion is important for accurately conveying the capabilities of the method. It is true that errors are not a liability for some applications, but knowing those error frequencies (and to some extent, understanding their origins) is essential for readers to determine whether REDI could be useful for their application or not. For these reasons, we believe it is best for the reader that we leave this section as is in the main text.

After sequencing 30,000 clones the authors observed only 3,316 unique sequences with perfect clones, which covers only 47% of their desired sequences. As the authors point out, doubling the number of clones sequenced would only give them an additional ~1000 oligos. So does that mean that REDI is only useful for applications that require only 47-61% of the designed oligos?

No, it does not. More iterations of the method can be performed for applications that require more of the library. In addition, as we now point out in lines 209-212, more uniform array-synthesized libraries and improved strategies for amplifying rare entities will improve the recovery of designed oligos. Finally, as pointed out by Reviewer #2, REDI could be used as a first pass, and then “missing” oligos could be acquired by some other means.

Major points

It is appreciated that the authors report cost and time required for this method, but given the fact that other similar methods exist it is not adequate to just compare chip-based synthesis vs. individual oligo synthesis (and even so, it seems a little disingenuous to say that all 3,316 oligos would be required to target 354 genomes).

We did not intend to suggest that 3,316 are required to target 354 genomes, but rather to compare the costs of REDI and commercial column-based synthesis for generating oligos at this scale. We have clarified that this applies to oligonucleotides to be used in any application on line 324. As noted above, we have also included a paragraph in the discussion comparing Dial-Out PCR and REDI (lines 326-342). While we do not include a direct cost comparison between these two methods (which is difficult without first-hand knowledge of current Dial-Out protocols), we do discuss the advantages and disadvantages of both methods.

Molecular Inversion Probes have been around for a long time, and people have synthesizing them on microarrays for years. What is it about their application that requires higher-quality probes?

As we state above, this application did not specifically require higher-quality probes, but we used it to develop the REDI method and to characterize its capabilities. We have now more clearly highlighted the established utility of probes made directly from microarrays, and have outlined some of the advantages of the REDI library that we observed on lines 343-361. As we point out in the response to reviewer #1, access to the individual probes allows for the creation

of tailored subcollections, which may be desirable for more in-depth profiling of low-abundance microbial species.

It would be more appropriate to compare REDI vs. similar existing methods like Dial-Out. They should also discuss how REDI addresses problems that are not already addressed by Dial-Out. While the authors mention three main features of REDI (capacity to generate equimolar libraries, ability to archive strains, and ability to directly deploy library members for cell-based interrogation of biological function), it is not clear that these are unique advantages of REDI. For example, Dial-Out could arguably be superior with regard to archiving, since desiccated DNA can be very cheaply and very stably stored for long durations on filter paper.

We agree that it is important to discuss the differences between REDI and dial-out PCR and (as noted above) have added a paragraph to the discussion (lines 326-342).

As the authors point out in Figure S1, they are likely observing PCR-dependent recombination of their oligos, resulting in "hybrid" oligos. Their method could be improved by doing qPCR of the oligo pools so that they can stop the PCR during exponential amplification (i.e., before the PCR plateaus). As is, it's hard to evaluate how much of a problem DNA synthesis is (they can't control that) vs. DNA amplification (they can control that).

We thank the reviewer for this suggestion, and have added a note at lines 178-184 to make this point.

I'm not totally convinced by Figure 2C. The authors do not report which of the original 7,051 probes from the array-synthesized library are observed in the 3,316 REDI probes. It would be useful to color the array-synthesized library probes that are also observed in the REDI library as one color, and the non-represented library probes as another color. Additionally, plotting the array-synthesized and REDI libraries' abundances on the same axes would make a clearer figure (i.e., there is 0 abundance in the REDI library for more than half of the total probes). Anyway, the calculations of uniformity should always be compared for the exact same subset of probes in both libraries (i.e., if you count a probe with an abundance of 10 in the array-synthesized library and it is not present in the REDI library, then you need to count the abundance in the REDI library as 0). It's not totally clear from the manuscript, but at least based on the figure, it doesn't appear that the authors did this.

We agree that plotting the CA and REDI libraries on the same axis is important for a robust comparison. Therefore, we plotted the 3,316 probes from the REDI library, and the same 3,316 probes for the Custom Array library (47% of this library) on the same axis in Fig.2C. We have tried coloring the probes differently as suggested by the reviewer, but there are too many data points to allow the different probe sets to be distinguished from one another. Similarly, plotting the ~3,700 probes that were never intended to be part of the REDI library did not improve the comparison of the two probe sets. We have also combined Fig.2C and Fig.2D (as suggested by Reviewer #1). To allow for a robust comparison, we randomly sampled the Illumina sequencing data so that each probe set was analyzed with the same number of reads. This is described in the methods (lines 621-628).

Figure 2E: Does this data represent any quantitative information about relative population frequencies of the different bacteria? Not only would that be more useful, it would also be a compelling reason that the probes would need to be uniformly represented in the REDI library. Also are the "other probes" with non-zero read counts false positives? Could the authors validate probe specificity empirically, and then remove these probes from the library for future use? Do the authors ever observe probes that don't work (i.e., false negatives)?

We included these results only to demonstrate that our REDI probe library was functional. This control experiment employed very similar concentrations of pure genomic DNA (see methods), and thus is not ideal for uncovering quantitative differences in populations. However, the observed read count range of probes targeting the same bacterium suggests that improvements beyond probe uniformity will need to be made to robustly detect quantitative differences. These efforts are ongoing in our lab.

The non-zero read counts of the “other probes” are indeed false positives. There are also four examples of false negatives. False positive and false negative probes could be easily removed from future libraries, as well as probes directed against highly-abundant species that dominate a sample, which is an important advantage of REDI. We now make this point in lines 160-167.

Minor points

Line 64: "accumulate" probably isn't the best word. It would be clearer to say that the probability of at least one synthesis error increases with each additional base added.

We agree and have made this change.

Figure 1 is confusing as currently laid out, making the method look like an iterative cycle. It would be helpful to spatially separate the first and last steps to avoid this confusion.

We have made this change to Figure 1.

Do the authors have any hypotheses for why the essential genes that are not related to respiration still seem to show less depletion in anoxic conditions? Is it an artifact, or a biological insight?

We think this observation is potentially interesting, but our experiments do not distinguish between two possibilities. We have added the following to the manuscript on lines 253-255: “This could reflect greater dosage sensitivity or alternatively, more effective gene repression during respiratory growth, which was ~2x slower than fermentative growth.”

"The CRISPRi collection 285 (Figure 3), from design through validation, took only a few months to complete and 286 required substantially less effort than what would be needed with traditional methods."-This claim needs to be qualified somehow. Some existing methods use microarray oligo pools without the added steps of REDI. It seems like better performance would be a better argument, but then the authors would need to actually compare performance of their REDI library vs. a non-sequence-verified library.

This claim actually refers to existing yeast collections for essential genes (i.e. temperature sensitive, DAmP, etc); we have qualified it accordingly (lines 314-316). A key attribute of our collection (as well as these other collections) is access to individual strains, providing researchers the option to study individual genes of interest or study phenotypes not amenable to pooled growth experiments. This is an important advantage over using microarray oligos to directly create plasmid-based pools for CRISPRi screens.

2nd Editorial Decision

10 January 2017

Thank you for sending us your revised manuscript. We have now heard back from reviewer #1 who was asked to evaluate your study. As you will see below, the reviewer thinks that the manuscript is improved and most issues have been satisfactorily addressed. However, reviewer #1 refers to the need to address two remaining minor points. As such, we would ask you to perform the requested changes in a minor revision.

REFeree REPORTS

Reviewer #1:

Thank you very much to the authors for addressing my critiques in detail. The revised manuscript is much improved and has addressed the majority of my points.

The authors have done a comprehensive and thorough job of creating, optimizing, characterizing

and demonstrating technical aspects of the REDI technique. As this stands technically, this part of the paper is well written and is an important contribution to the field.

The authors write a number of interesting approaches that could be enabled by the REDI technique, such as repooling of an optimized or targeted strain library, phenotypic screening of a larger (i.e. greater than a few dozen) number of strains. I think it would be ideal to demonstrate a biological application beyond the library resource generation study described in this manuscript, but perhaps this could be explored in future studies.

Minor points

- 1) In Fig. 2, the 3 steps for the MIP approach could be labeled to increase clarity.
- 2) While the additional bioinformatics tool information is greatly appreciated, the authors still have not added information as to quality filtering and trimming of raw sequencing data. This information should be added to increase reproducibility of their pipeline.

2nd Revision - authors' response

12 January 2017

Reviewer #1:

Thank you very much to the authors for addressing my critiques in detail. The revised manuscript is much improved and has addressed the majority of my points.

The authors have done a comprehensive and thorough job of creating, optimizing, characterizing and demonstrating technical aspects of the REDI technique. As this stands technically, this part of the paper is well written and is an important contribution to the field.

The authors write a number of interesting approaches that could be enabled by the REDI technique, such as repooling of an optimized or targeted strain library, phenotypic screening of a larger (i.e. greater than a few dozen) number of strains. I think it would be ideal to demonstrate a biological application beyond the library resource generation study described in this manuscript, but perhaps this could be explored in future studies.

We would like to thank the reviewer for his/her constructive advice and for taking the time to carefully review our manuscript.

Minor points

- 1) In Fig. 2, the 3 steps for the MIP approach could be labeled to increase clarity.

We have added text to Fig.2A to better describe the approach and increase clarity.

- 2) While the additional bioinformatics tool information is greatly appreciated, the authors still have not added information as to quality filtering and trimming of raw sequencing data. This information should be added to increase reproducibility of their pipeline.

We did not filter the raw sequencing data using quality scores or trimming. To limit the effects of DNA sequencing errors, we did the following:

In some of the analyses (e.g. Fig.2B, 2C, 3B), we required that Read1 and Read2 were perfect matches of each other, thereby greatly diminishing the chance of a sequencing artifact influencing the read counts. This is already stated on lines 623-626 of the manuscript
For the analysis of diploid recombinants produced by REDI, we first binned reads requiring perfect matches to the desired indexing sequences (i.e. the 6nt indexing barcodes identifying the plate and the 26nt REDI barcodes identifying plate position). The sequencing experiments are designed such that each bin (representing a single colony) will contain many sequence reads (typically >100). As sequencing errors are, in general, predicted to occur somewhat randomly throughout the length of the read, the 'real' sequence (i.e. the incorporated oligo) is expected to be the dominant sequence among the multiple reads in each bin. We have clarified

this in the methods section (lines 605-611) of the manuscript (also see below; modified text is in red).

Briefly, sequencing reads were binned according to short (i.e. six nucleotide) indexing barcodes included in both the forward and reverse sequencing primers (index pairs were unique to each plate of diploid recombinants), and, then, by the 26 bp barcode (unique to each barcoder strain and, thus, colony position in the arrayed plate of recombinants). These steps required sequences to perfectly match the designed barcodes (all other sequences were excluded from further analysis). The exogenous DNA sequence in each colony was then identified as the most commonly observed sequence, between the common priming regions, in each set of binned reads. This sequence was compared to the designed molecular probe or guide RNA libraries and sequences perfectly matching a designed sequence were prioritized for selection. We further refined this list by de-prioritizing clones where the perfect match sequence was supported by fewer than 50 reads. In some cases, we also filtered out diploid recombinants having multiple different sequences that perfectly matched multiple different designed sequences (likely indicating robotic picking of a 'mixed colony').

3rd Editorial Decision

12 January 2017

Thank you for sending us your revised manuscript. We are now satisfied with the modifications made and I am pleased to inform you that your paper has been accepted for publication.

Corresponding Author Name: Bob St.Onge
Journal Submitted to: Molecular Systems Biology
Manuscript Number: MSB-16-7233